# SENSITIVITY ANALYSIS FOR DIFFUSION MODELS

## ABSTRACT

Training a diffusion model approximates a map from a data distribution $\rho$ to the optimal score function $s_t$ for that distribution. Can we differentiate this map? If we could, then we could predict how the score, and ultimately the model's samples, would change under small perturbations to the training set before committing to costly retraining. We give a closed-form procedure for computing this map's directional derivatives, relying only on black-box access to a pre-trained score model and its derivatives with respect to its inputs. We extend this result to estimate the sensitivity of a diffusion model's samples to additive perturbations of its target measure, with runtime comparable to sampling from a diffusion model and computing log-likelihoods along the sample path. Our method is robust to numerical and approximation error, and the resulting sensitivities correlate with changes in an image diffusion model's samples after retraining and fine-tuning.

## 1 INTRODUCTION

Diffusion models form a powerful class of generative models that allow users to generate images of nearly any subject in nearly any style in just a few keystrokes. However, this flexibility also allows diffusion models to engage in legally fraught behavior, such as generating images that mimic an artist's style. This has put diffusion models at the center of recent litigation[1] alleging that they facilitate copyright and trademark infringement. Understanding and mitigating the causes of this behavior have therefore become pressing challenges as businesses seek to integrate diffusion models into their consumer offerings.

Diffusion models generate images by iteratively transforming Gaussian noise using the *score function* of a distribution over noisy images, which is learned in practice from a large set of training images. Since the learned score is, in principle, determined by the training images, a natural strategy for understanding a diffusion model's behavior is to study how it depends on the training data. A sensible framework for this task should be able to answer questions such as: "What would the score function be if a sample were added or removed from the training set?" and "What would a generated image have looked like if a sample were added or removed from the training set?"

This work introduces a principled framework for answering such questions about diffusion models in the perturbative regime, where one considers infinitesimal changes in a model's training distribution. Because a diffusion model's output depends on the score function, which is itself determined by the training distribution, the core of our framework is a tractable closed-form expression for the directional derivatives of a score function with respect to the training distribution. This *sensitivity analysis* measures how the score function changes as a probability measure is up- or down-weighted in the training distribution; when this measure is a Dirac point mass, we obtain an exact expression for the *influence function* of the score. Crucially, our sensitivity analysis requires only black-box access to a pre-trained score function, and it does not require any knowledge of its training data or training procedure.

Using the adjoint method, we extend our sensitivity analysis for score functions to obtain schemes for computing the sensitivity of a diffusion model's samples to perturbations in the training data. These enable us to predict how a generated image would change if a collection of samples were added or removed from the training set. We demonstrate our method's robustness to a variety of sources of numerical error and show that its predictions are correlated with changes in a diffusion model's samples after retraining and after fine-tuning.

---

[1] *Andersen v. Stability AI Ltd.*, U.S. District Court, Northern District of California (2024).

## 2 RELATED WORK

**Influence functions.** Influence functions (Hampel, 1974) linearly approximate the change in a statistical estimator in response to infinitesimally upweighting a single training sample. Koh & Liang (2017) introduced influence functions to deep learning as a method for estimating the change in a neural network's parameters in response to perturbing its training set. Influence functions for generic optima of a training loss require the user to compute a costly inverse Hessian-vector product. Previous work (Guo et al., 2021; Schioppa et al., 2022) responds to this challenge by developing efficient approximations to this operation. In addition to this difficulty, influence functions assume that the learned model parameters minimize a strictly convex loss. This assumption is violated for neural networks, and Basu et al. (2020) find that in practice, influence functions for deep learning are brittle to network hyperparameters. Kwon et al. (2024); Mlodozeniec et al. (2025) introduce influence approximations that are specially adapted to generative models, including diffusion models, but these works follow Koh & Liang (2017) in estimating the influence of training samples on the learned network weights. In contrast, our sensitivity analysis uses the structure of diffusion models to directly compute the influence of training samples on the value of the score function and on model samples.

**Data attribution for diffusion models.** An emerging literature develops *data attribution* methods for diffusion models, which seek to estimate the impact of training samples on model outputs. Georgiev et al. (2023) use TRAK (Park et al., 2023), a gradient-based data attribution method developed primarily for supervised learning, to compute a per-example attribution score for a diffusion model's training data. This score estimates the change in the model's training loss induced by adding a particular sample to the training set. Following Park et al. (2023), they measure their method's effectiveness using the *linear datamodeling score*, which measures the rank correlation between their attribution score and actual training loss values attained by retrained models. Zheng et al. (2024) observe that one can improve upon the method from Georgiev et al. (2023) by computing their attribution scores using the gradients of the "wrong" model output function. Mlodozeniec et al. (2025) introduce an efficient approximation to the denoising loss Hessian and use it to estimate influence functions for attributing several proxies for model log-probabilities. Lin et al. (2025) propose attributing the KL divergence between the model distribution before and after deleting a training sample. Li et al. (2025) perform gradient-based data attribution using learnable weights for gradients with respect to different parameter groups. Whereas these methods estimate the impact of training samples on scalar quantities such as the training loss or proxies for log-probabilities, our sensitivity analysis estimates the effect of perturbations in the target distribution on the values of the score function and on model samples.

## 3 METHOD

In this section, we first observe that a diffusion model defines a map from its training distribution $\rho$ to a score function $s_t$ and show how to tractably compute its directional derivatives. Using this result, we estimate how a diffusion model's samples change when its training distribution is perturbed.

### 3.1 PRELIMINARIES

*Diffusion models* sample from a target distribution $\rho$ by drawing samples from a Gaussian base distribution $\mathcal{N}(0, I)$ and flowing them through a possibly noisy velocity field $v_t$ from $t = t_0$ to $t = t_1$. This yields a curve of probability distributions $\{\rho_t : t \in [t_0, t_1]\}$ for which $\rho_t$ is the marginal distribution of the random variable $Z_t := \alpha_t X_1 + \sigma_t \epsilon$. Here, $X_1 \sim \rho$, $\epsilon \sim \mathcal{N}(0, I)$, and $\alpha_t$ and $\sigma_t$ are scale and noise schedules, respectively. These schedules are chosen so that at $t = t_0$, the samples have a Gaussian distribution, and at $t = t_1$, the samples are distributed according to $\rho$.

A diffusion model's velocity field $v_t$ depends on $\rho$ through the *score function* $s_t(z) := \nabla \log \rho_t(z)$ of $\rho_t$, which one learns in practice by solving a *score-matching* problem (Hyvärinen & Dayan, 2005). If one does not impose any restrictions on the hypothesis class, the optimal solution to this problem is in fact available in closed form (Miyasawa, 1961), yielding a vector field pointing from $z$ toward a distance-weighted average of rescaled samples $\alpha_t x$ from the target distribution $\rho$. Solving the score-matching problem therefore maps a *measure* $\rho$ to a *function* $s_t$, which is fully determined by $\rho$ and the scale and noise schedules.

We would like to estimate how the outputs of a diffusion model would change in response to perturbations of the training data. These outputs depend on the training data only through the velocity field $v_t$ and, in turn, through the score function $s_t$. We will therefore begin by introducing a tractable closed-form expression for the directional derivatives of the map from $\rho$ to $s_t$, which will describe how $s_t$ responds to additive perturbations of the target distribution $\rho$.

## 3.2 SENSITIVITY ANALYSIS FOR SCORE FUNCTIONS

Solving the score-matching problem maps a target distribution $\rho$ to a score function $s_t$. To understand how $s_t$ changes in response to small perturbations of $\rho$, intuitively one would like to differentiate $s_t$ with respect to the probability measure $\rho$. However, it is not obvious how to compute this derivative in practice. In this section, we present a tractable formula for such a derivative with respect to *additive* perturbations of $\rho$. This class of perturbations includes many cases of interest, such as the addition of new samples and the removal of existing samples from the training set.

Suppose that $\rho^\eta := (1 - \eta)\rho + \eta\nu$ is a *mixture* of two probability measures $\rho$ and $\nu$ supported on $\mathbb{R}^d$, and let $s_t^\eta : \mathbb{R}^d \to \mathbb{R}^d$ be the score function of a diffusion model with target distribution $\rho^\eta$ at time $t$. Differentiating $s_t^\eta$ with respect to $\eta$ and evaluating this derivative at $\bar{\eta}$ yields a *function* $g_t^{\bar{\eta}} : \mathbb{R}^d \to \mathbb{R}^d$ describing how $s_t^\eta$ changes as one infinitesimally upweights $\nu$ given initial weight $\bar{\eta}$.

The case $\bar{\eta} = 0$ is of particular interest. For example, if $\bar{\eta} = 0$, then $g_t^{\bar{\eta}}$ describes how a score function trained on $\rho$ would vary as one introduces samples from $\nu$. On the other hand, to approximate how the score function of a diffusion model trained on $\rho$ would change in response to *removing* training data lying in some region $\Omega \subseteq \mathbb{R}^d$, one would define $\nu := \rho_\Omega$, where $\rho_\Omega$ is the restriction of $\rho$ to $\Omega$, and consider $-g_t^{\bar{\eta}}$ evaluated at $\bar{\eta} = 0$.

**Our key result is the following theorem**, which provides a tractable closed-form expression for $g_t^0$:

**Theorem 3.1 (Sensitivity analysis for score functions)** *For $\eta \in [0, 1]$, let $\rho^\eta := (1 - \eta)\rho + \eta\nu$ be a mixture of probability measures $\rho$ and $\nu$ with compact support on $\mathbb{R}^d$. Let $\rho_t^\eta : \mathbb{R}^d \to \mathbb{R}$ and $s_t^\eta : \mathbb{R}^d \to \mathbb{R}^d$ be the density and score function, resp., of a diffusion model with target distribution $\rho^\eta$ at time $t \in [t_0, t_1)$. Then the Fréchet derivative in $L^2(\mathbb{R}^d, \rho_t^\eta)$ of the map $T_t(\eta) : \eta \mapsto s_t^\eta$ evaluated at $\bar{\eta} = 0$ is the function $g_t^0 : \mathbb{R}^d \to \mathbb{R}^d$ defined by the formula:*

$$g_t^0(z) = \frac{\nu_t(z)}{\rho_t(z)} \left( s_t^\nu(z) - s_t^\rho(z) \right), \tag{1}$$

*where $\nu_t(z), \rho_t(z)$ are the respective densities and $s_t^\nu(z), s_t^\rho(z)$ the respective scores at time $t$ of diffusion models with target measures $\nu, \rho$. Moreover, for every fixed $z \in \mathbb{R}^d$, the pointwise derivative satisfies $\frac{\partial}{\partial \eta} s_t^\eta(z)\Big|_{\eta=0} = g_t^0(z)$ without any assumptions on the support of $\mu$ or $\nu$.*

We prove this result in Appendix B.1. Whereas score-matching defines a map from a measure $\rho$ to the unique optimal score function $s_t$ of a diffusion model with $\rho$ as its target, Equation 1 now provides a formula for the *directional derivative* of this map in the direction of $\nu - \rho$. In Figure 1, we depict an instance of this directional derivative $g_t^0$ when $\rho$ is supported on a curve in 2D and $\nu$ is a Gaussian measure centered just off the curve. $g_t^0$ is a vector field pointing away from the support of $\rho$ and towards the support of $\nu$; on account of the $\nu_t(z)/\rho_t(z)$ scaling factor, $\|g_t^0(z)\|_2$ is large at points $z$ that are closer to the support of $\nu$ than to the support of $\rho$.

In a typical use case, $\rho_t$ is the distribution at time $t$ of a diffusion model trained on $\rho$, and $\nu = \frac{1}{K} \sum_{k=1}^K \delta_{x_k}$ is the empirical distribution on $K$ samples $x_k$ that one wishes to add or remove from $\rho$; if $K = 1$, we recover the *influence function* of $T_t$ (Hampel, 1974). In this setting, we may use Equation 1 to compute $g_t(z)$ given *only blackbox access* to the score function $s_t^\rho(z)$ and the $K$ samples $x_k$. The density $\rho_t(z)$ of the diffusion model can be computed from its score using the continuous change of variables (CCoV) formula (Song et al., 2021), and since $\nu_t$ is a mixture of Gaussians when $\nu$ is an empirical distri-

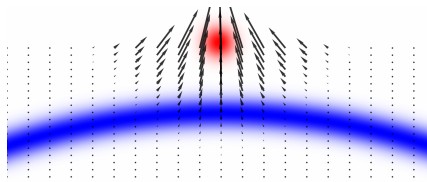

Figure 1: The score sensitivity $g_t^0$ is a vector field pointing away from the support of $\rho$ and towards the support of $\nu$.

bution, its density and score function can be computed in closed form in $O(dK)$ time or efficiently

approximated using techniques from Karppa et al. (2022); Scarvelis et al. (2025). For perturbation sets $S = \{x_k\}_{k=1}^K$ of moderate size, the cost of evaluating Equation 1 is dominated by the cost of the density computations $\rho_t(z)$ using the CCoV formula. Appendix A provides further background on the CCoV formula and score and density computations for mixtures of Gaussians.

Theorem 3.1 shows how to tractably estimate the response of a pre-trained score function to additive perturbations in its target distribution. However, in practice, we are typically interested in how the *samples* generated by a diffusion model would change in response to perturbing its target distribution. Because samples are obtained by solving an ODE or SDE determined by the score function, Equation 1 should provide enough information to estimate the sensitivity of model samples to additive perturbations of $\rho$. We show this to be the case in the following section, using the adjoint method to obtain an analogous perturbation formula for a diffusion model's samples.

### 3.3 SENSITIVITY ANALYSIS FOR MODEL SAMPLES

Diffusion models generate samples from their target distribution $\rho$ by solving a stochastic differential equation (SDE) or an ordinary differential equation (ODE) whose *drift* or *velocity field*, respectively, depend on the score $s_t^\eta$. Because this dependence is typically simple, often consisting of an affine transformation of $s_t^\eta$, it is easy to differentiate the drift or velocity field with respect to $\eta$ given Equation 1. In this section, we exploit this fact to compute the sensitivity of a diffusion model's samples to additive perturbations of the target distribution.

**ODE sampling.** We begin with the simpler case of ODE sampling. Song et al. (2021) show that one may sample a diffusion model by solving a *probability flow ODE* (PF-ODE), whose initial condition is drawn from the Gaussian base distribution: $\frac{dz_t}{dt} = v_t^\eta(z_t)$ with $z_0 \sim \mathcal{N}(0, I)$. Because the Lipschitz constant of $s_t^\eta$ – and consequently $v_t^\eta$ – may blow up as $t \to t_1$, we follow a common convention from the theory of diffusion models and truncate integration of $v_t^\eta$ at some $\tilde{t}_1 < t_1$ (De Bortoli, 2022). This convention aligns with typical diffusion model sampling schemes, which return samples at some time $\tilde{t}_1$ slightly earlier than the theoretical sampling interval endpoint $t_1$.

If one further assumes that the target distributions $\mu, \nu$ are compactly supported on $\mathbb{R}^d$, then a typical $v_t^\eta(z)$ will be globally Lipschitz for $z \in \mathbb{R}^d$ and $t \in [t_0, \tilde{t}_1]$. Khalil (2002, Theorem 3.2) then shows that there exists a unique solution to the PF-ODE for any initial condition $z_0 \in \mathbb{R}^d$. This allows us to define a *solution map* $\Phi_s^\eta(z_0) : \mathbb{R}^d \to \mathbb{R}^d$ that maps an initial condition $z_0 \in \mathbb{R}^d$ to the unique solution at time $s \in [t_0, \tilde{t}_1]$ of the initial value problem (IVP) defined by $v_t^\eta$. Intuitively, $\Phi_s^\eta(z_0)$ maps an initial noise sample $z_0 \sim \mathcal{N}(0, I)$ to the sample's position at time $s$ along the diffusion model's sample path; at time $s = \tilde{t}_1$, this is simply a model sample.

We are interested in the derivative $\frac{d}{d\eta}\Phi_{\tilde{t}_1}^\eta(z_0)$ for fixed initial conditions $z_0$, which describes how the model sample $\Phi_{\tilde{t}_1}^\eta(z_0)$ generated from the Gaussian sample $z_0$ varies as one perturbs the target distribution $\mu$ in the direction of $\nu$. Khalil (2002, Section 3.3) shows that under certain regularity conditions, this derivative solves an ODE known as the *sensitivity equation*. Defining $\psi_s := \frac{d}{d\eta}\Phi_s^\eta(z_0)$ and letting $z_s := \Phi_s^\eta(z_0)$ for $s \in [t_0, \tilde{t}_1]$ be a solution path for the PF-ODE, this equation is:

$$\frac{d}{ds}\psi_s = \frac{d}{d\eta}v_s^\eta(z_s) + J_z[v_s^\eta](z_s)\psi_s, \tag{2}$$

where the initial condition is $\psi_{t_0} = 0$ and $J_z[v_s^\eta](z_s)$ denotes the spatial Jacobian of $v_s^\eta$ evaluated at $z_s$. A solution $\psi_{\tilde{t}_1} = \frac{d}{d\eta}\Phi_{\tilde{t}_1}^\eta(z_0)$ to Equation 2, which we will call a *sample sensitivity*, approximates the change in a sample $z_{\tilde{t}_1} = \Phi_{\tilde{t}_1}^\eta(z_0)$ in response to additive perturbations of the target distribution $\rho$. Figure 2 depicts a solution to Equation 2 when $\rho$ is supported on a curve in 2D and $\nu$ is a Gaussian measure centered just off the curve.

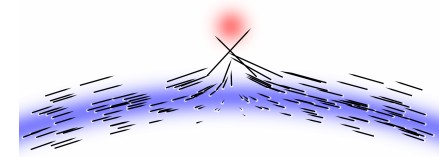

Figure 2: A solution to Equation 2 approximates the change in a diffusion model's samples as the target distribution $\rho$ is perturbed in the direction of $\nu$.

Crucially, one may solve Equation 2 given black-box access to the score function $s_t^\eta$ and its spatial derivatives. To estimate how a sample $\Phi_s^\eta(z_0)$ generated from initial noise $z_0$ would change in response to perturbing $\rho$, one should (1) compute a sample path $z_t$ and

---

**Algorithm 1** Sample sensitivity analysis

---

**Require:** Score model $s_t^\rho$; perturbation measure $\nu$; initial noise $z_{t_0} \sim \rho_{t_0}$; time interval $[t_0, \tilde{t}_1]$
  1: $\psi_{t_0} \leftarrow 0$ ▷ Initialize sample sensitivity to 0
  2: $z_t \leftarrow \text{SamplePath}(s_t, z_{t_0})$ ▷ Compute $z_t$ via ODE/SDE sampling
  3: $v_t \leftarrow \mathcal{T}(s_t)$ ▷ Transform score to PF-ODE velocity field
  4: $\log \rho_t(z_t) \leftarrow \log \rho_{t_0}(z_{t_0}) - \int_{t_0}^{t} \nabla \cdot v_s(z_s) \mathrm{d}s$ ▷ Compute log-densities via CCoV formula (3)
  5: **if** $\nu$ is an empirical measure on $\{x_i\}_{i=1}^N$ **then**
  6:     Compute $\nu_t(z_t)$ and $s_t^\nu(z_t)$ as closed-form density (9) and score (10) of a Gaussian mixture
  7: **else if** $\nu$ is parametrized by a neural score model **then**
  8:     Compute $\nu_t(z_t)$ via the CCoV formula (3)
  9: **end if**
 10: $\left.\frac{\mathrm{d}}{\mathrm{d}\eta} s_t^\eta(z_t)\right|_{\eta=0} \leftarrow \frac{\nu_t(z_t)}{\rho_t(z_t)} \left(s_t^\nu(z_t) - s_t^\rho(z_t)\right)$ ▷ Compute score sensitivities via Eq. 1

 11: $\psi_t \leftarrow \psi_{t_0} + \int_{t_0}^{t} \left( \left.\frac{\mathrm{d}}{\mathrm{d}\eta} v_s^\eta(z_s)\right|_{\eta=0} + J_z[v_s](z_s)\psi_s \right) \mathrm{d}s$ ▷ Compute sample sensitivity path via Eq. 2
 12: **return** $\psi_t$ ▷ Return sample sensitivity path

---

model densities $\rho_t(z_t)$ by jointly integrating the PF-ODE and the CCoV formula, (2) evaluate Equation 1 along the sample path, which also entails computing the density and score of the perturbation measure $\nu$, and (3) integrate Equation 2, using autograd to compute the spatial Jacobian-vector products $J_z[v_s^\eta](z_s)\psi_s$. We summarize this procedure in Algorithm 1. When $\nu$ is an empirical measure over $K$ samples, our sample sensitivity analysis has time complexity $O((hP + dK)T)$, where $h$ is the number of noise samples used in Hutchinson's trace estimator, $P$ is an architecture-dependent constant measuring the cost of evaluating the score network $s_t^\rho$, and $T$ is the number of time steps in the ODE discretization. Its space complexity is $O(dT)$, where $d$ is the ambient dimension.

**SDE sampling.** In practice, it is more common to sample a diffusion model by solving an SDE $\mathrm{d}z_t = f_t^\eta(z_t)\mathrm{d}t + g_t \mathrm{d}W_t$, where $W_t$ denotes a Wiener process on $\mathbb{R}^d$. Only the drift coefficient $f_t^\eta$ depends on the score function $s_t^\eta$ and consequently on $\eta$; conversely, the *diffusion coefficient* $g_t$ is independent of $\eta$. Kunita (2019, Theorem 3.3.2) provides an analogous sensitivity analysis for the solution of an SDE whose coefficients depend on a parameter. Suppose an SDE has a unique solution and let $\Gamma_{s,\omega}^\eta : \mathbb{R}^d \to \mathbb{R}^d$ be the solution map sending an initial condition $z_0$ to the SDE's solution at time $s \in [t_0, t_1]$ for a fixed realization $\omega$ of the Wiener process. Kunita (2019, Theorem 3.3.2) shows that under certain regularity conditions, which are satisfied for typical drifts if one truncates the integration at $\tilde{t}_1 < t_1$, $\frac{\mathrm{d}}{\mathrm{d}\eta}\Gamma_{s,\omega}^\eta(z_0)$ also satisfies some SDE for almost all $\omega$. Moreover, when the diffusion coefficient $g_t$ is independent of $\eta$ and the spatial variable, this differential equation is, in fact, deterministic and coincides with the sensitivity analysis for ODE sampling from Equation 2. We may therefore use Equation 2 to approximate the change in a diffusion model's SDE samples in response to perturbations of its target distribution. In practice, one follows the recipe from the previous section on ODE sampling, but replaces the ODE sample path $z_t$ with an SDE sample path $z_t$.

## 4 EXPERIMENTS

This section empirically validates our sensitivity analysis for diffusion models. We begin by studying the effect of approximation error using synthetic data with known scores and densities. We then experiment with neural diffusion models trained on image datasets and show that our sample sensitivities correlate with changes in model samples after retraining and fine-tuning. We conclude by studying key statistics of our sample sensitivities for models trained on image datasets.

### 4.1 FIRST-ORDER APPROXIMATION FOR PERTURBED MODEL SAMPLES

A solution $\psi_{\tilde{t}_1} = \frac{\mathrm{d}}{\mathrm{d}\eta}\Phi_{\tilde{t}_1}^\eta(z_0)|_{\eta=0}$ to Equation 2 estimates how a diffusion model's samples change under an additive perturbation of the target distribution, yielding a first-order approximation $\Phi_{\tilde{t}_1}^{\bar{\eta}}(z_0) \approx \Phi_{\tilde{t}_1}^0(z_0) + \bar{\eta}\frac{\mathrm{d}}{\mathrm{d}\eta}\Phi_{\tilde{t}_1}^\eta(z_0)|_{\eta=0}$ that converges at rate $o(\bar{\eta})$ by Taylor's theorem. However, in practice, error from numerically solving Equation 2 can degrade the accuracy of this approximation

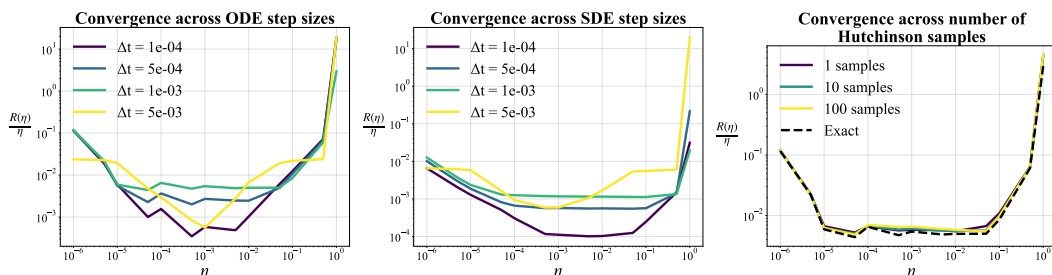

Figure 3: The approximation error of our first-order approximation to a perturbed model's samples decays at rate $o(\bar{\eta})$ for a variety of ODE step sizes (left) and SDE step sizes (center), and this rate is robust to noise from Hutchinson's estimator (right).

for realistic step sizes, and evaluating Equation 1 requires computing $\rho_t$ using the CCoV formula, which introduces additional noise through Hutchinson's estimator for $\mathrm{div}(v_t)$. In this section, we use synthetic data with exact scores and densities to study the effects of ODE integration error and density estimation error on the convergence of our linear approximation to perturbed model samples. Appendix F.1.1 provides implementation details.

**Effect of step size.** In this experiment, we study how step size affects the convergence of our first-order approximation to the perturbed model's samples when solving Equation 2 with a forward Euler scheme. We choose the initial target measure $\rho$ to be an equally weighted mixture of well-separated Gaussians in $\mathbb{R}^{100}$. This multimodal, high-dimensional target distribution simulates some of the mathematical pathologies of real-world data; in particular, its score function is nearly discontinuous near Voronoi boundaries between the mixture means. For every $t$, the corresponding $\rho_t$ remains a Gaussian mixture with closed-form score and density, allowing us to isolate the effect of ODE/SDE discretization error, density approximation error, and score approximation error on our sensitivity analysis. Because we perturb $\rho$ toward a second Gaussian $\nu$, the perturbed target $\rho^{\bar{\eta}} = (1 - \bar{\eta})\rho + \bar{\eta}\nu$ also remains a Gaussian mixture. Since the first-order approximation uses only pointwise derivatives, $\rho$ and $\nu$ need not have compact support.

We generate sample paths $z_t$ for $\rho_t$ and $\rho_t^{\bar{\eta}}$ by numerically integrating the PF-ODE and the VP-SDE using forward Euler and Euler–Maruyama with several step sizes, and exactly compute $\rho_t(z_t)$ along each path. We then integrate Equation 2 with the same forward Euler scheme to compute the sensitivities $\frac{\mathrm{d}}{\mathrm{d}\eta}\Phi_{t_1}^{\eta}(z_0)|_{\eta=0}$ of initial samples to perturbations toward $\nu$. Taylor's theorem implies that a first-order approximation's error $R(\bar{\eta})$ is $o(\bar{\eta})$, so we compute $R(\bar{\eta})$ for $\bar{\eta} \in [0, 1]$ and verify this rate in practice.

The left and center panels of Figure 3 depicts the results of this experiment for ODE and SDE sample paths. Linearly approximating samples from a perturbed target $\rho^{\bar{\eta}}$ using our sample sensitivity analysis (2) is accurate within $o(\bar{\eta})$ for a variety of step sizes and $\bar{\eta}$. For very small values of $\bar{\eta}$, $R(\bar{\eta})/\bar{\eta}$ plateaus and begins to increase again. This reflects a noise floor in the accuracy of model samples, which are themselves computed by numerically integrating an ODE or SDE.

**Effect of Hutchinson's estimator.** In the previous experiment, we computed the model densities $\rho_t(z)$ in (1) exactly by choosing a target distribution for which $\rho_t$ has a closed form. In practice, however, diffusion models approximate $\nabla \log \rho_t(z)$ with a neural network, from which we may recover densities via the CCoV formula $\frac{\mathrm{d}\log\rho_t(z_t)}{\mathrm{d}t} = -\mathrm{tr}(J_{z_t}[v_t](z_t))$. To avoid forming a large Jacobian, one uses Hutchinson's estimator $\mathrm{tr}(A) = \mathbb{E}[\epsilon^{\top} A \epsilon]$, whose accuracy depends on the number of $\epsilon$ samples. To study this estimator's impact on the accuracy of our first-order approximation, we repeat the previous experiment with step size $10^{-3}$ but estimate $\rho_t(z)$ using the CCoV formula with a varying number of $\epsilon$ samples. The right panel of Figure 3 plots the scaled remainders $R(\bar{\eta})/\bar{\eta}$ when using exact densities (dashed line) and Hutchinson's estimator with 1, 10, and 100 $\epsilon$ samples. Our method's convergence rate is robust to noise in Hutchinson's estimator, with even a single $\epsilon$ achieving nearly the same approximation error as the exact densities for all but the largest $\eta$.

## 4.2 STABILITY OF SAMPLE SENSITIVITY UNDER SCORE APPROXIMATION ERROR

Section 4.1 showed that one may approximate perturbed samples $\Phi_{\tilde{t}_1}^{\bar{\eta}}(z_0)$ using our sensitivity analysis formula (2) and recover the expected $o(\bar{\eta})$ convergence rate despite errors from numerical integration and Hutchinson's estimator. To isolate the effects of these errors, we used the exact score $\nabla \log \rho_t$ of the mixture of Gaussians $\rho_t$ throughout our computations. In practice, however, one typically *learns* this score function by training a neural network on a score-matching objective, introducing additional error. Here, we show that our sample sensitivity analysis (2) is stable to approximation error in the score function.

We take $\rho$ to be a mixture of well-separated Gaussians on $\mathbb{R}^{10}$ and perturb it toward a Gaussian measure $\nu$. Rather than evaluating the score of $\rho_t$ in closed form as in Section 4.1, we train a neural network to approximate it. We fix $z_0 \sim \rho_0$ and compute the exact and approximate models' sample sensitivities every 1000 training steps. We discretize all ODEs with forward Euler and estimate $\rho_t(z)$ using Hutchinson's estimator. At each step, we compute the median correlation between the exact and approximate sample sensitivities and compare it to the score-matching loss. Appendix F.1.2 gives additional implementation details.

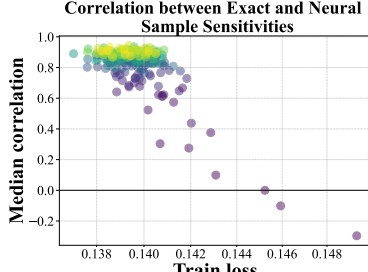

Figure 4: The correlation between sample sensitivities for an exact diffusion model and its neural approximation rises rapidly as the training loss falls. Points are colored from purple to yellow according to the training step.

Figure 4 shows the relationship between the training loss and the median correlation. Points are colored by training step; we omit the first two early measurements where the loss is very large. The correlations rise rapidly as the loss decreases, indicating that our sample sensitivity analysis is robust to score-approximation error and remains informative even when the exact score is replaced by a learned approximation. In the next section, we build on this observation by showing that our sample sensitivities predict the direction of change in a diffusion model's samples after retraining on a perturbed target distribution.

### 4.3 PREDICTING CHANGES IN MODEL SAMPLES VIA SAMPLE SENSITIVITY ANALYSIS

**Predicting change in model samples after retraining.** In the previous section, we used synthetic data from a mixture of Gaussians to study the robustness of our sensitivity analysis to various sources of numerical error. In practice, diffusion models are trained on large datasets of images, with training often stopped well before convergence to prevent memorization (Favero et al., 2025). In this section, we demonstrate that our sample sensitivities $\psi_{\tilde{t}_1}$ are correlated with differences between an image diffusion model's samples before and after retraining on a perturbed target distribution. We experiment with UNet-based diffusion models trained on a mixture of the MNIST and Typography-MNIST (TMNIST) datasets (Magre & Brown, 2022) and on the CelebA dataset (Liu et al., 2015).

For each dataset, we train a base model and a perturbed model whose target distribution $\rho^{\bar{\eta}}$ is a mixture of the base model's target distribution and the empirical measure on a set of new samples $S$. We employ mixture weights $\bar{\eta} = 0.1$ and $1 - \bar{\eta} = 0.9$, resp. For our MNIST experiment, the new samples $S$ are drawn from TMNIST, and for our CelebA experiment, $S$ consists of samples with a large CLIP score for "a photo of an old man." We integrate the PF-ODE to obtain model samples from $\rho^0$ and $\rho^{\bar{\eta}}$, and also integrate Equation 2 with the perturbation measure $\nu$ set to the empirical distribution over $S$ to estimate the sensitivity of the base model's samples to upweighting $S$. We compare the sample sensitivities $\frac{\mathrm{d}}{\mathrm{d}\eta}\Phi_{\tilde{t}_1}^{\eta}(z_0)|_{\eta=0}$ to the difference $\Phi_{\tilde{t}_1}^{\bar{\eta}}(z_0) - \Phi_{\tilde{t}_1}^0(z_0)$ between PF-ODE samples from the perturbed and base model given the same initial noise. This measures how much our sample sensitivity analysis predicts actual changes in model samples after retraining on the perturbed target distribution $\rho^{\bar{\eta}}$. Appendix F.2 provides further implementation details.

Figure 5 depicts histograms of the correlations between our sample sensitivities and the actual change in model samples. As a baseline, we also compute the entropic optimal transport (OT) coupling (Cuturi, 2013) between the base model samples $\Phi_{\tilde{t}_1}^0(z_0)$ and the target distribution for the perturbed model and use the resulting transport rays as predicted directions of change in the model samples after retraining. These transport rays are line segments connecting the base model samples to their coupled samples from the perturbed model's target distribution under the entropic OT coupling, providing a robust, *training-free* baseline for how the base model outputs might respond

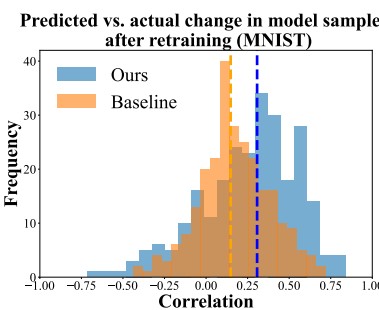 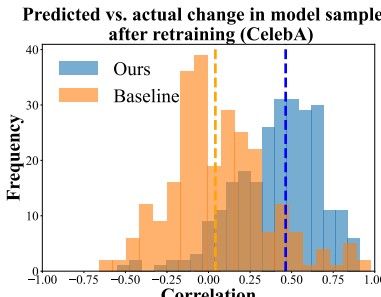

Figure 5: Correlations between predicted and actual change in model samples after retraining on a perturbed dataset. Our sample sensitivity analysis (blue) outperforms an optimal transport baseline (orange), achieving a median correlation (dashed blue line) of 0.46 on CelebA and 0.31 on MNIST.

to perturbations of the training set. Our sample sensitivity scores correlate with actual changes in model samples after retraining on $\rho^{\bar{\eta}}$ and substantially outperform the OT baseline, achieving a median correlation of 0.46 on CelebA and 0.31 on MNIST, compared 0.04 and 0.15, resp., for the OT baseline. We conjecture that our method's especially strong performance on CelebA reflects the structure of the dataset, which contains face images with large regions of relatively uniform pixel values. Our sensitivity analysis achieves high correlations by accurately predicting the pixel-level changes in these regions, whose near-uniformity simplifies the task.

In this setting, we do not expect our sensitivity analysis to perfectly predict how model samples respond to perturbations of the training set. One reason is that neural score models trained on large datasets of images are typically *not* optimal solutions to the score-matching problem; in fact, Pidstrigach (2022) shows that any score-based generative model that generalizes must incur unbounded approximation error. While Section 4.2 shows that our sensitivity analysis is stable to reasonable score approximation error, this approximation error is large for typical neural diffusion models. Furthermore, diffusion models trained via gradient descent are *stable* to small perturbations in their training set (Favero et al., 2025), so the experiments in this section necessarily operate outside the small-perturbation regime where our sensitivity analysis is most predictive.

**Predicting change in model samples after fine-tuning.** The previous experiment shows that our sample sensitivities $\frac{\mathrm{d}}{\mathrm{d}\eta}\Phi^{\eta}_{\bar{t}_1}(z_0)|_{\eta=0}$ correlate with changes in model samples after retraining on a perturbed target distribution. We will now show that our sample sensitivities are more strongly predictive of changes in model samples after *fine-tuning* on new training samples $S$. We use the same base models and the same $S$ as in the previous experiment, but fine-tune on $S$ rather than retraining from scratch on the mixture distribution $\rho^{\bar{\eta}}$.

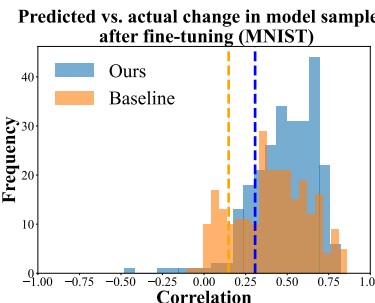 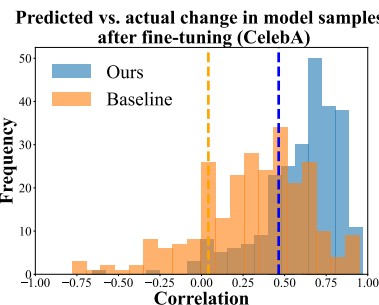

Figure 6: Our sample sensitivities (blue) are correlated with changes in model samples after fine-tuning, and continue to outperform an optimal transport baseline (orange).

We depict histograms of the correlations between our sample sensitivities and actual change in model samples after fine-tuning in Figure 6. We use the same entropic OT baseline as in the previous experiment, but compute transport rays between the base model samples and the samples $S$ on which we fine-tuned. Both our sample sensitivities and the OT baseline are better correlated with actual

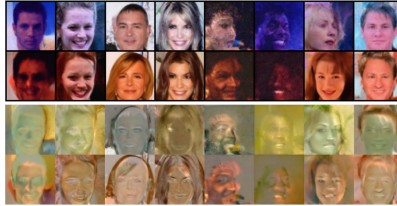

Figure 7: Top rows show samples from the original models; second rows show samples after fine-tuning. The third rows display our sample sensitivities, which predict changes in model samples after fine-tuning (fourth row). On the left, the model is trained on MNIST and fine-tuned on TMNIST; on the right, the model is trained on CelebA and fine-tuned on a subset of faces with large CLIP score for "a photo of an old man."

change in model samples after fine-tuning, but our method continues to outperform the baseline, achieving a median correlation of 0.66 on CelebA and 0.51 on MNIST, compared to 0.34 and 0.42, resp., for the baseline. We visually compare our sample sensitivities to actual changes in model samples after fine-tuning in Figure 7, which shows that our sample sensitivity analysis can provide coarse predictions of how a diffusion model's samples might change after fine-tuning. We provide further illustrations of our sample sensitivity analysis in Appendix E.

### 4.4 EMPIRICAL PROPERTIES OF SAMPLE SENSITIVITIES

We now study certain empirical properties of our sample sensitivities. We begin by computing the sensitivity of four samples from the previous section's CelebA model to each of its training samples. Each sample sensitivity in these experiments is a vector in $\mathbb{R}^d$ that predicts how a model sample would change in response to infinitesimally upweighting a single training sample; we refer to their magnitudes as each training sample's *influence score* with respect to the model sample. We present several notable findings below.

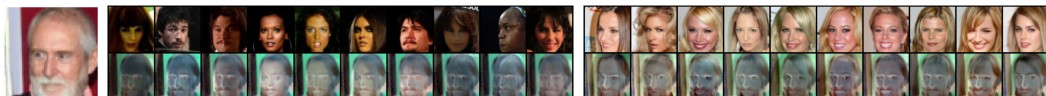

Figure 8: Most and least influential training samples (center, right, resp.) for the model sample on the left, with the corresponding sensitivities on the bottom row.

**Influence scores are correlated with $L_2$ distances.** In Figure 8, we visualize the top-10 and bottom-10 influential training samples for the model sample depicted on the left of the figure. (See Figure 12 in Appendix C for the remaining model samples.) These outliers are characterized by large regions of homogeneously bright or dark pixels, suggesting that influence scores may be correlated with the $L_2$ distance between the model sample and each training sample, which is sensitive to large differences in per-pixel intensity.

We validate this conjecture by regressing the training samples' influence scores on their $L_2$ distance from each model sample. These regressions' $r^2$ values are substantial, ranging from 0.80 to 0.91 for the model samples in this experiment, indicating that the distance from a training sample to a model sample predicts its influence score. However, there is useful information in this regression's residuals, which capture how much more or less influential a training sample is than one would expect based on its distance to the model sample. Figure 9 shows the top-10 and bottom-10 training samples according to their residual influence scores. (See Figure 13 in Appendix C for the remaining model samples.) The training samples with the greatest residual influence tend to share the model sample's pose but vary substantially in their facial expression, whereas the samples with the least residual influence possess outlier features such as hats or glasses.

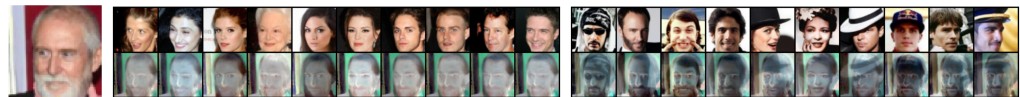

Figure 9: Training samples with the largest and smallest residual influence scores (center, right, resp.) for the model sample on the left, with the corresponding sensitivities on the bottom row.

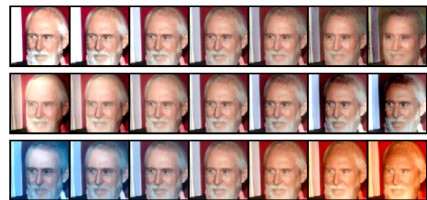

**Sample sensitivities lie in low-dimensional subspaces.**
We now compute the singular value decomposition
(SVD) of the matrix of a fixed model sample's sensi-
tivities to each training sample and find that these sam-
ple sensitivities are nearly low-dimensional, with $93\%$ of
their variance explained by the first 10 singular directions.
Moreover, these directions are often interpretable. In Fig-
ure 10, we depict perturbation rays of the model sample
of the form $x + \alpha u$, where $x$ is a model sample and $u$
is a right-singular vector of its sensitivities with respect
to each training sample. The first singular vector in the
figure appears to control beard density, the second con-

Figure 10: Singular vectors of a model
sample's sensitivities to its training
samples yield interpretable perturba-
tions. (Top to bottom: beard, lighting,
color temperature.)

trols the direction of the light source, and the third controls color temperature. (See Figure 14 in
Appendix D for raw singular vectors.) These results suggest that training set perturbations influ-
ence model samples along only a few degrees of freedom, a phenomenon which may arise from the
low-dimensional structure of the data manifold.

**Cross-class sensitivities are small.** We finally leverage the
availability of class labels in the CIFAR-10 dataset to investi-
gate how training samples from one class might influence sam-
ples from another. For each class $C_i$ in CIFAR-10, we compute
the average magnitude of the sensitivity of model samples from
class $C_i$ to samples from every other class $C_j$. Figure 11 depicts
a heatmap whose $(i, j)$-th entry represents the average sensitivity
of samples from class $C_i$ to training samples from class $C_j$. This
heatmap is nearly diagonal, showing that model samples are pri-
marily influenced by training samples from their own class. This
may reflect the CIFAR-10 dataset's union-of-manifolds structure,
which has previously been observed by Brown et al. (2023).

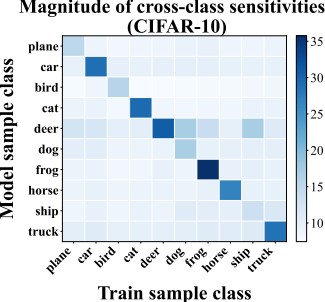

Figure 11: Cross-class sensitiv-
ities are small for CIFAR-10.

## 5  DISCUSSION

Understanding a diffusion model's dependence on its training data is a critical challenge in machine
learning. In general, one would expect the relationship between a large model and its training data
to be complex and difficult to estimate. This paper shows that it is not only possible to compute
directional derivatives of the map from a training distribution $\rho$ to its optimal score function $s_t$,
but that this computation is (a) surprisingly cheap, costing roughly as much as sampling a model
and computing log-probabilities along the sample path, and (b) requires only black-box access to
the score function. One can then leverage this simple formula to estimate how a diffusion model's
samples change in response to perturbations to its target distribution before retraining or fine-tuning
on new data. We propose several future directions for this line of work.

Throughout this paper, we perturbed a diffusion model's target measure with empirical measures
over finite samples. This need not be the case: Our score sensitivity formula (1) holds for any
compactly-supported perturbation measure $\nu$, and it can be implemented in practice for any sequence
of measures $\nu_t$ provided we can access their scores and densities. For instance, $\nu_t$ can be a second
diffusion model, in which case Equation 1 resembles the formula for classifier-free guidance (CFG)
(Ho & Salimans, 2022) with time- and spatially-varying weights. Future work might interpret CFG
in light of our sensitivity analysis and design new guidance schedules based on this formula.

By composing a model's ODE sampling solution map with a text-conditioned classifier and apply-
ing our sensitivity formulas, one might also use our method to estimate how the likelihood that a
diffusion model's samples match a prompt changes as one perturbs the training set. This would
allow users to attribute a model's qualitative behavior to subsets of training samples and use this
information to curate the training set to steer a diffusion model's behavior in a particular direction.

Finally, Kadkhodaie et al. (2024) find empirically that a diffusion model's sampling map is often
insensitive to changes in its training set, and Favero et al. (2025) clarify that this behavior is con-
trolled by the number of training iterations, with models becoming increasingly sensitive to dataset
perturbations throughout training. Our sample sensitivity formula (2) *quantifies* this dependence and
may serve as a valuable tool for future work on generalization in diffusion models.

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

## A    EXTENDED PRELIMINARIES

For the sake of completeness, this appendix summarizes known results that have been used elsewhere in this manuscript.

**The continuous change of variables (CCoV) formula.**    Let $\rho_t$ be a continuous family of time-indexed probability measures on $\mathbb{R}^d$ whose densities we denote by $\rho_t(z)$. Suppose that samples $z_t \sim \rho_t$ evolve according to the ODE $\frac{\mathrm{d}}{\mathrm{d}t}z_t = f(z_t, t)$, where $f : \mathbb{R}^d \times \mathbb{R} \to \mathbb{R}^d$ is uniformly Lipschitz in $z_t$ and continuous in $t$. Then the log-density $\log \rho_t(z_t)$ of $z_t$ evolves according to the *continuous change of variables formula*:

$$\frac{\mathrm{d}}{\mathrm{d}t} \log \rho_t(z_t) = -\nabla \cdot f(z_t, t), \tag{3}$$

where $\nabla \cdot f(z_t, t) = \mathrm{tr}(J_{z_t} f(z_t, t))$ denotes the divergence of $f(z_t, t)$ with respect to its first argument. This result appears in Chen et al. (2018, Theorem 1). In the setting of diffusion models, $f(z_t, t)$ is the PF-ODE velocity field (Song et al., 2021), which we denote by $v_t$ in this manuscript.

In practice, it is often prohibitive to compute $\nabla \cdot f(z_t, t) = \mathrm{tr}(J_{z_t} f(z_t, t))$ by explicitly forming the $d \times d$ Jacobian matrix $J_{z_t} f(z_t, t)$. To mitigate the computational burden, one typically employs *Hutchinson's trace estimator* (Hutchinson, 1990) $\mathrm{tr}(A) = \mathbb{E}_\epsilon[\epsilon^\top A \epsilon]$, which holds for any random variable $\epsilon$ with mean 0 and identity covariance. The key advantage of this estimator is that one can compute Monte Carlo approximations $\mathrm{tr}(J_{z_t} f(z_t, t)) \approx \frac{1}{h} \sum_{i=1}^{h} \epsilon_i^\top J_{z_t} f(z_t, t)) \epsilon_i$ using only Jacobian-vector products (JVPs), whose time complexity via automatic differentiation is at most $\frac{5}{2} \times$ the complexity of evaluating $f$ (Griewank & Walther, 2008, Chapter 3).

**Entropy-regularized optimal transport.**    The results in this section are drawn from Peyré & Cuturi (2020). Let $\mu$ and $\nu$ be two probability measures on $\mathbb{R}^d$. *Monge's problem* seeks a pushforward $T : \mathbb{R}^d \to \mathbb{R}^d$ of $\mu$ onto $\nu$ that minimizes the average distance $\|x - T(x)\|_2$ between coupled units of probability mass:

$$\min_{T : \nu = T_\# \mu} \int \|x - T(x)\|_2 \mathrm{d}\mu(x). \tag{4}$$

This map has appealing geometric properties: for instance, *transport rays* $T(x) - x$ do not cross on their interior (Caffarelli et al., 2000). As Monge's problem may not have a solution, one typically relaxes this problem to a search for a *coupling*: A probability measure $\pi$ on $\mathbb{R}^d \times \mathbb{R}^d$ whose marginals are $\mu$ and $\nu$. This relaxation yields the well-known *Kantorovich problem*:

$$W_1(\mu, \nu) := \min_{\pi \in \Pi(\mu, \nu)} \int \|x - y\|_2 \mathrm{d}\pi(x, y), \tag{5}$$

where $\Pi$ denotes the set of probability measures whose marginals are $\mu, \nu$. The optimal value of this problem is the *1-Wasserstein distance* between $\mu$ and $\nu$. When $\mu, \nu$ are discrete measures supported on $\{x_i\}_{i=1}^N, \{y_j\}_{j=1}^M$, Equation 5 reduces to:

$$W_1(\mu, \nu) := \min_{T \in \Pi(\mu,\nu)} \sum_{i=1}^N \sum_{j=1}^M T_{ij} \|x_i - y_j\|_2, \qquad (6)$$

where $T \in \Pi(\mu, \nu)$ is now a $n \times m$ matrix whose row and column sums are equal to $\mu$ and $\nu$, respectively. When $N = M$, there exists a solution to the linear program (6) that also solves the Monge problem (4). One may interpolate between $\mu$ and $\nu$ by moving samples from the support of $\mu$ along transport rays, which are straight line segments connecting $x$ to $T(x)$.

While Equation 6 is defined by a linear program which can be solved in principle, doing so is costly, especially for large-scale problems in machine learning and graphics. To mitigate this cost, Cuturi (2013) proposes to regularize Equation 6 with an entropy term:

$$\text{Sinkhorn}(\mu, \nu) := \min_{T \in \Pi(\mu,\nu)} \sum_{i=1}^N \sum_{j=1}^M T_{ij} \|x_i - y_j\|_2 + \epsilon \sum_{i=1}^N \sum_{j=1}^M T_{ij} \log T_{ij}. \qquad (7)$$

This approximation enables the use of *Sinkhorn's algorithm* to solve the entropy-regularized optimal transport problem in quadratic time. In addition to its computational benefits, entropy regularization is often desirable for high-dimensional machine learning problems with noisy data. We use Sinkhorn's algorithm to approximate the optimal coupling between the base model samples and samples from the perturbed model's target distribution to compute our baseline in Section 4.3.

**ODE sensitivity equation.** The results in this section are drawn from Khalil (2002, Section 3.3). Given a function $f(z, t, \lambda)$, suppose that $z_t$ satisfies the ODE $\frac{d}{dt} z_t = f(z_t^\lambda, t, \lambda)$, where $\lambda \in \mathbb{R}^p$ is a parameter that may be interpreted as a control vector. Suppose also that $f$ is continuous in all its arguments and is continuously differentiable with respect to $z_t$ and $\lambda$ for all $t$. Let $\lambda_0$ be a parameter for which the initial value problem $\frac{d}{dt} z_t^{\lambda_0} = f(z_t, t, \lambda_0)$ with initial condition $z_0$ has a unique solution over some interval $[t_0, t_1]$. Then the solution path $z_t^{\lambda_0}$ is differentiable with respect to $\lambda$ near $\lambda_0$, and this derivative $S_t := \frac{d}{d\lambda} z_t^\lambda \big|_{\lambda=\lambda_0}$ satisfies the *sensitivity equation*:

$$\frac{d}{dt} S_t = \frac{\partial f}{\partial z}(z_t^{\lambda_0}, t, \lambda_0) \cdot S_t + \frac{\partial f}{\partial \lambda}(z_t^{\lambda_0}, t, \lambda_0). \qquad (8)$$

We use this ODE sensitivity equation to derive our sample sensitivity analysis in Section 3.3.

**Score functions and density functions for mixtures of Gaussians.** In practice, the perturbation measure $\nu$ is typically the empirical measure on $K$ samples $x_k \in \mathbb{R}^d$ that one wishes to add or remove from $\rho$. In this case, $\nu_t$ is a mixture of isotropic Gaussians for all $t \in [t_0, t_1]$:

$$\nu_t(z) = \frac{1}{K} \sum_{k=1}^K \mathcal{N}(z; \alpha_t x_k; \sigma_t^2 I), \qquad (9)$$

where $\alpha_t$ and $\sigma_t$ are the scaling and noise schedules, respectively. One may therefore compute the exact density of $\nu_t$ with time complexity $O(dK)$. The score of $\nu_t$ is also available in closed form:

$$\nabla \log \nu_t(z) = \frac{1}{\sigma_t^2} (k_t(z) - z), \qquad (10)$$

$$\text{where } k_t(z) = \sum_{k=1}^K \text{softmax} \left( -\frac{\|z - \alpha_t X\|^2}{2\sigma_t^2} \right)_i \alpha_t x_k, \qquad (11)$$

in which we let $\|z - \alpha_t X\|^2$ denote the vector whose $k$-th entry is $\|z - \alpha_t x_k\|^2$. This score can also be computed exactly in $O(dK)$ time.

In settings where $K$ is large, these density and score computations may be prohibitive. Fortunately, the large sums in the density and score computations are well-structured: For small $\sigma_t$, both sums are dominated by the term involving the $x_k$ nearest to $z$, and for large $\sigma_t$ the scalar terms in each sum are approximately uniform. Karppa et al. (2022) show how to exploit this structure to efficiently approximate densities of the form (9) using approximate nearest-neighbor queries, and Scarvelis et al. (2025) use similar techniques to efficiently approximate score functions of the form (10).

# B    PROOFS

## B.1    PROOF OF THEOREM 3.1

We will prove this theorem in two parts. We will first show that $\left.\frac{\partial}{\partial\eta}s_t^\eta(z)\right|_{\eta=0} = g_t(z)$ at any fixed $z \in \mathbb{R}^d$. This shows that $g_t(z)$ is the pointwise derivative of $s_t^\eta$ evaluated at $\eta = 0$ at any $z \in \mathbb{R}^d$. We will then extend this pointwise argument to the space of functions by using the dominated convergence theorem (DCT) to prove that $g_t$ is the Fréchet derivative in $L^2(\mathbb{R}^d, \rho_t^0)$ of the map $T(\eta) : \eta \mapsto s_t^\eta$.

### B.1.1    $g_t(z)$ IS THE POINTWISE DERIVATIVE OF $s_t^\eta$ AT $\eta = 0$

In this part of the proof, we will rely heavily on Mlodozeniec et al. (2025, Lemma 1). A version of their lemma adapted to our setting states the following:

**Lemma B.1** *Let $\mathcal{L} : \mathbb{R} \times \mathbb{R}^d \to \mathbb{R}$ be a $C^2$ function of additive form $\mathcal{L}(\eta, s_z) := \mathcal{L}_1(s_z) + \eta\mathcal{L}_2(s_z)$, and suppose that the map $s_z \mapsto \mathcal{L}(\eta, s_z)$ is strictly convex for all $\eta \in \mathbb{R}$. Fix $\bar{\eta}$ and choose $s_z^*$ such that $\frac{\partial \mathcal{L}}{\partial s_z}(\bar{\eta}, s_z^*) = 0$. Then, by applying the implicit function theorem to $\frac{\partial \mathcal{L}}{\partial s_z}$, one obtains an open interval $(-\delta, \delta) \subseteq \mathbb{R}$ containing $\bar{\eta}$ and a unique function $\phi : (-\delta, \delta) \to \mathbb{R}^d$ such that $\phi(\bar{\eta}) = s_z^*$ and such that for all $\eta \in (-\delta, \delta)$, $\phi(\eta)$ is the unique minimizer of $s_z \mapsto \mathcal{L}(\eta, s_z)$. Moreover, $\phi$ is $C^1$ with the following derivative:*

$$\frac{\partial}{\partial\eta}\phi(\eta) = -\left[\frac{\partial^2 \mathcal{L}}{\partial s_z^2}(\eta, \phi(\eta))\right]^{-1} \frac{\partial \mathcal{L}_2}{\partial s_z}(\phi(\eta)). \tag{12}$$

To apply this lemma, we will first show that the score function $s_t(z)$ of the marginal distribution of $Z_t = \alpha_t X_1 + \sigma_t \epsilon$ can be characterized pointwise at any $z \in \mathbb{R}^d$ as the minimizer of a score-matching objective. For the sake of simplicity, we will assume a constant scale schedule $\alpha_t \equiv 1$; our argument can be easily adapted to arbitrary scale schedules at the cost of additional notation.

Let $s_t(z) : \mathbb{R}^d \to \mathbb{R}^d$ be the score function for some distribution $\rho_t := \rho * \mathcal{N}(0, \sigma_t^2 I)$, where $\rho$ is a target distribution on $\mathbb{R}^d$. Kadkhodaie et al. (2024, Eqs. 14, 19) and the variational characterization of conditional expectation imply that this score function has the following pointwise variational characterization:

$$\begin{aligned}
s_t(z) &= \nabla \log \rho_t(z) \\
&= \int_{\mathbb{R}^d} \left(\frac{x-z}{\sigma_t^2}\right) p(x|z)\mathrm{d}x \\
&= \underset{s(z)}{\mathrm{argmin}} \int \frac{1}{2}\left\|\frac{x-z}{\sigma_t^2} - s(z)\right\|^2 p(x|z)\mathrm{d}x,
\end{aligned}$$

where $p(x|z)$ is the conditional distribution of $x$ given $z \sim \rho_t$. While $p(x|z)$ is intractable a priori, $p(z|x) \sim \mathcal{N}(x, \sigma_t^2 I)$ is Gaussian, so we rewrite $p(x|z)$ in this integral using Bayes' theorem:

$$s_t(z) = \underset{s_z \in \mathbb{R}^d}{\operatorname{argmin}} \int \frac{1}{2} \left\| \frac{x-z}{\sigma_t^2} - s_z \right\|^2 p(x|z) \mathrm{d}x$$

$$= \underset{s_z \in \mathbb{R}^d}{\operatorname{argmin}} \int \frac{1}{2} \left\| \frac{x-z}{\sigma_t^2} - s_z \right\|^2 \frac{\mathcal{N}(z; x, \sigma_t^2 I)}{\rho_t(z)} \rho(x) \mathrm{d}x$$

$$= \underset{s_z \in \mathbb{R}^d}{\operatorname{argmin}} \underset{x \sim \rho}{\mathbb{E}} \left[ \frac{\mathcal{N}(z; x, \sigma_t^2 I)}{\rho_t(z)} \frac{1}{2} \left\| \frac{x-z}{\sigma_t^2} - s_z \right\|^2 \right]. \tag{SM}$$

Here, we use $\mathcal{N}(z; x, \sigma_t^2 I)$ to denote the density of a Gaussian distribution with mean $x$ and co-variance $\sigma_t^2 I$ evaluated at $z \in \mathbb{R}^d$. This provides a pointwise definition of the score $s_t(z)$ of $\rho_t$ evaluated at $z \in \mathbb{R}^d$ as the minimizer of the score-matching problem $(SM)$. In particular, applying this argument to the target distribution $\rho^\eta$ shows that:

$$s_t^\eta(z) = \underset{s_z \in \mathbb{R}^d}{\operatorname{argmin}} \underset{x \sim \rho^\eta}{\mathbb{E}} \left[ \frac{\mathcal{N}(z; x, \sigma_t^2 I)}{\rho_t^\eta(z)} \frac{1}{2} \| \frac{x-z}{\sigma_t^2} - s_z \|^2 \right].$$

Now, define the following objective functions, in which we take $z \in \mathbb{R}^d$ to be fixed:

$$\mathcal{L}_\rho(s_z) = \underset{x \sim \rho}{\mathbb{E}} \left[ \frac{\mathcal{N}(z; x, \sigma_t^2 I)}{\rho_t^\eta(z)} \frac{1}{2} \left\| \frac{x-z}{\sigma_t^2} - s_z \right\|^2 \right]$$

and

$$\mathcal{L}_\nu(s_z) = \underset{x \sim \nu}{\mathbb{E}} \left[ \frac{\mathcal{N}(z; x, \sigma_t^2 I)}{\rho_t^\eta(z)} \frac{1}{2} \left\| \frac{x-z}{\sigma_t^2} - s_z \right\|^2 \right].$$

Using these two objectives, we can define an objective $\mathcal{L}(\eta, s_z) := \underbrace{\mathcal{L}_\rho(s_z)}_{:=\mathcal{L}_1} + \eta \underbrace{(\mathcal{L}_\nu(s_z) - \mathcal{L}_\rho(s_z))}_{:=\mathcal{L}_2}$

whose minimizer is $s_t^\eta(z)$. This objective is in the additive form prescribed by Lemma B.1, which will put us in position to apply the lemma once we verify that its remaining hypotheses are satisfied.

To this end, note that $\mathcal{L}$ is clearly a $C^2$ function of the prescribed additive form. Furthermore, as we will see below via a Hessian computation, the map $s_z \mapsto \mathcal{L}(\eta, s_z)$ is strictly convex for all $\eta \in \mathbb{R}$ and for all $z \in \mathbb{R}^d$. Fixing a point $(\bar{\eta}, s_z^*) = (\bar{\eta}, s_t^\eta(z))$ yields a critical point of $\mathcal{L}$ with respect to $s_z$, which puts us in position to apply Lemma B.1.

Lemma B.1 gives us a function $\phi(\eta)$ defined on an open set containing $\bar{\eta}$ that maps $\eta$ to the unique minimizer $s_t^\eta(z)$ of $\mathcal{L}(\eta, s_z)$. Crucially, it gives us a formula for the derivative $\frac{\partial}{\partial \eta} \phi(\eta)$, which involves the derivative $\frac{\partial \mathcal{L}_2}{\partial s_z}(\phi(\eta))$ and the Hessian $\frac{\partial^2 \mathcal{L}}{\partial s_z^2}(\eta, \phi(\eta))$. We will compute each of these terms separately.

We begin by computing the derivative $\frac{\partial \mathcal{L}_2}{\partial s_z}(\phi(\eta))$. We have

$$\frac{\partial \mathcal{L}_2}{\partial s_z}(\phi(\eta)) = \frac{\partial}{\partial s_z} \left[ \mathbb{E}_{x \sim \nu} \left[ \frac{\mathcal{N}(z; x, \sigma_t^2 I)}{\rho_t^\eta(z)} \frac{1}{2} \left\| \frac{x-z}{\sigma_t^2} - s_z \right\|^2 \right] - \mathbb{E}_{x \sim \rho} \left[ \frac{\mathcal{N}(z; x, \sigma_t^2 I)}{\rho_t^\eta(z)} \frac{1}{2} \left\| \frac{x-z}{\sigma_t^2} - s_z \right\|^2 \right] \right] \Bigg|_{s_t^\eta(z)}$$

$$= \frac{\partial}{\partial s_z} \mathbb{E}_{x \sim (\nu - \rho)} \left[ \frac{\mathcal{N}(z; x, \sigma_t^2 I)}{\rho_t^\eta(z)} \frac{1}{2} \left\| \frac{x-z}{\sigma_t^2} - s_z \right\|^2 \right] \Bigg|_{s_t^\eta(z)}$$

$$= \mathbb{E}_{x \sim (\nu - \rho)} \left[ \frac{\mathcal{N}(z; x, \sigma_t^2 I)}{\rho_t^\eta(z)} \frac{\partial}{\partial s_z} \frac{1}{2} \left\| \frac{x-z}{\sigma_t^2} - s_z \right\|^2 \right] \Bigg|_{s_t^\eta(z)}$$

$$= \mathbb{E}_{x \sim \nu} \left[ -\frac{\mathcal{N}(z; x, \sigma_t^2 I)}{\rho_t^\eta(z)} \left( \frac{x-z}{\sigma_t^2} - s_z \right) \right] \Bigg|_{s_t^\eta(z)} - \mathbb{E}_{x \sim \rho} \left[ -\frac{\mathcal{N}(z; x, \sigma_t^2 I)}{\rho_t^\eta(z)} \left( \frac{x-z}{\sigma_t^2} - s_z \right) \right] \Bigg|_{s_t^\eta(z)}$$

$$= \mathbb{E}_{x \sim \rho} \left[ \frac{\mathcal{N}(z; x, \sigma_t^2 I)}{\rho_t^\eta(z)} \left( \frac{x-z}{\sigma_t^2} - s_t^\eta(z) \right) \right] - \mathbb{E}_{x \sim \nu} \left[ \frac{\mathcal{N}(z; x, \sigma_t^2 I)}{\rho_t^\eta(z)} \left( \frac{x-z}{\sigma_t^2} - s_t^\eta(z) \right) \right]$$

We now rewrite each expectation in the last line in terms of the scores of $\rho_t$ and $\nu_t$. To rewrite the first expectation, we pull out the factor of $\frac{1}{\rho_t^\eta(z)}$, which does not depend on $x$, and multiply by $1 = \frac{\rho_t(z)}{\rho_t(z)}$ to obtain the following:

$$\mathbb{E}_{x \sim \rho} \left[ \frac{\mathcal{N}(z; x, \sigma_t^2 I)}{\rho_t^\eta(z)} \left( \frac{x-z}{\sigma_t^2} - s_t^\eta(z) \right) \right] = \frac{\rho_t(z)}{\rho_t^\eta(z)} \mathbb{E}_{x \sim \rho} \left[ \frac{\mathcal{N}(z; x, \sigma_t^2 I)}{\rho_t(z)} \left( \frac{x-z}{\sigma_t^2} - s_t^\eta(z) \right) \right]$$

$$= \frac{\rho_t(z)}{\rho_t^\eta(z)} \left( \underbrace{\mathbb{E}_{x \sim \rho} \left[ \frac{\mathcal{N}(z; x, \sigma_t^2 I)}{\rho_t(z)} \left( \frac{x-z}{\sigma_t^2} \right) \right]}_{= s_t^\rho(z)} - \underbrace{\mathbb{E}_{x \sim \rho} \left[ \frac{\mathcal{N}(z; x, \sigma_t^2 I)}{\rho_t(z)} \right]}_{= 1} s_t^\eta(z) \right)$$

$$= \frac{\rho_t(z)}{\rho_t^\eta(z)} \left( s_t^\rho(z) - s_t^\eta(z) \right).$$

Analogous reasoning allows us to conclude that

$$\mathbb{E}_{x \sim \nu} \left[ \frac{\mathcal{N}(z; x, \sigma_t^2 I)}{\rho_t^\eta(z)} \left( \frac{x-z}{\sigma_t^2} - s_t^\eta(z) \right) \right] = \frac{\nu_t(z)}{\rho_t^\eta(z)} \left( s_t^\nu(z) - s_t^\eta(z) \right),$$

and putting these together, we obtain

$$\frac{\partial \mathcal{L}_2}{\partial s_z}(\phi(\eta)) = \frac{\rho_t(z)}{\rho_t^\eta(z)} s_t^\rho(z) - \frac{\nu_t(z)}{\rho_t^\eta(z)} s_t^\nu(z) + \left( \frac{\nu_t(z) - \rho_t(z)}{\rho_t^\eta(z)} \right) s_t^\eta(z). \tag{13}$$

We now compute the Hessian term $\frac{\partial^2 \mathcal{L}}{\partial s_z^2}(\eta, \phi(\eta))$. Note that $\frac{\partial^2 \mathcal{L}}{\partial s_z^2} = \frac{\partial^2}{\partial s_z^2} \mathcal{L}_\rho + \eta(\frac{\partial^2}{\partial s_z^2} \mathcal{L}_\nu - \frac{\partial^2}{\partial s_z^2} \mathcal{L}_\rho)$, and that we have already computed the relevant first derivatives:

$$\frac{\partial}{\partial s_z} \mathcal{L}_\rho = \mathbb{E}_{x \sim \rho} \left[ -\frac{\mathcal{N}(z; x, \sigma_t^2 I)}{\rho_t^\eta(z)} \left( \frac{x-z}{\sigma_t^2} - s_z \right) \right]$$

and

$$\frac{\partial}{\partial s_z} \mathcal{L}_\nu = \mathbb{E}_{x \sim \nu} \left[ -\frac{\mathcal{N}(z; x, \sigma_t^2 I)}{\rho_t^\eta(z)} \left( \frac{x-z}{\sigma_t^2} - s_z \right) \right].$$

Differentiating again and simplifying, we see that

$$\frac{\partial^2}{\partial s_z^2} \mathcal{L}_\rho = \frac{\rho_t(z)}{\rho_t^\eta(z)} I$$

and

$$\frac{\partial^2}{\partial s_z^2} \mathcal{L}_\nu = \frac{\nu_t(z)}{\rho_t^\eta(z)} I.$$

Combining these and noting that $\rho_t^\eta = (1 - \eta)\rho_t + \eta\nu_t$, we conclude that $\frac{\partial^2 \mathcal{L}}{\partial s_z^2}(\eta, s_z) = I$ for all $\eta \in \mathbb{R}$ and for all $s_z$. In particular, the map $s_z \mapsto \mathcal{L}(\eta, s_z)$ is strictly convex for all $\eta \in [0, 1]$ as required by Lemma B.1.

We finally substitute these first and second derivatives into Equation 12 to obtain:

$$\frac{\partial}{\partial \eta} \phi(\eta) = - \left[ \frac{\partial^2 \mathcal{L}}{\partial s_z^2}(\eta, \phi(\eta)) \right]^{-1} \frac{\partial \mathcal{L}_2}{\partial s_z}(\phi(\eta))$$

$$= -[I]^{-1} \left( \frac{\rho_t(z)}{\rho_t^\eta(z)} s_t^\rho(z) - \frac{\nu_t(z)}{\rho_t^\eta(z)} s_t^\nu(z) + \left( \frac{\nu_t(z) - \rho_t(z)}{\rho_t^\eta(z)} \right) s_t^\eta(z) \right)$$

$$= \frac{\nu_t(z)}{\rho_t^\eta(z)} s_t^\nu(z) - \frac{\rho_t(z)}{\rho_t^\eta(z)} s_t^\rho(z) + \left( \frac{\rho_t(z) - \nu_t(z)}{\rho_t^\eta(z)} \right) s_t^\eta(z).$$

In particular, if $\eta = 0$, then $\rho_t^\eta(z) = \rho_t(z)$ and this simplifies to:

$$\frac{\partial}{\partial \eta} \phi(\eta) = \frac{\nu_t(z)}{\rho_t(z)} (s_t^\nu(z) - s_t^\rho(z)) =: g_t(z).$$

This completes the first part of the proof.

### B.1.2 $g_t$ IS THE FRÉCHET DERIVATIVE OF $T_t(\eta) : \eta \mapsto s_t^\eta$ AT $\eta = 0$

We now extend this pointwise argument to the space of functions. Consider the map $T_t : \mathbb{R} \to L^2(\mathbb{R}^d, \rho_t^\eta)$ that maps $\eta$ to $s_t^\eta$. We will show that $g_t$ is the Fréchet derivative of $T_t$ at $\eta = 0$ for any $t \in [t_0, t_1]$. To do so, we need to show that for any $t \in [t_0, t_1]$,

$$\lim_{h \to 0} \left\| \frac{s_t^h - s_t^0}{h} - g_t \right\|_{L^2(\mathbb{R}^d, \rho_t^0)} = 0.$$

The previous section shows that $g_t(z)$ is the *pointwise* derivative of $s_t^\eta$ with respect to $\eta$ at $\eta = 0$. This means that for any $z \in \mathbb{R}^d, t \in [t_0, t_1]$,

$$\frac{s_t^h(z) - s_t^0(z)}{h} \to g_t(z).$$

Hence $\frac{s_t^h - s_t^0}{h}$ converges pointwise to $g_t$ for all $t \in [t_0, t_1]$. We will use the dominated convergence theorem (DCT) to lift this pointwise convergence to $L^2(\mathbb{R}^d, \rho_t^0)$ convergence. Define the following function:

$$F_h(z; t) := \frac{s_t^h(z) - s_t^0(z)}{h}$$

We need to show that there exists some real-valued function $G(z; t) \in L^2(\mathbb{R}^d, \rho_t^0)$ such that $\|F_h(z; t)\|_2 \leq G(z; t)$ uniformly in $h$ for all $z, t$. To this end, note that by the mean value theorem, there exists some $\theta \in [0, 1]$ such that:

$$\|F_h(z;t)\|_2 = \left\|\frac{s_t^h(z) - s_t^0(z)}{h}\right\|_2$$

$$\leq \left\|\frac{\partial}{\partial \eta} s_t^\eta(z)\Big|_{\eta=\theta h}\right\|_2$$

$$= \left\|\frac{\rho_t(z)}{\rho_t^{\theta h}(z)}\left(s_t^\rho(z) - s_t^{\theta h}(z)\right) - \frac{\nu_t(z)}{\rho_t^{\theta h}(z)}\left(s_t^\nu(z) - s_t^{\theta h}(z)\right)\right\|_2,$$

where the last line follows from a rearrangement of Equation 13. We can further simplify this bound to eliminate the dependence on $h$. First, note that $\rho_t^{\theta h} = (1 - \theta h)\rho_t + \theta h \nu_t$, so that for $h \leq \frac{1}{2}$, we have

$$\frac{1}{\rho_t^{\theta h}(z)} = \frac{1}{(1 - \theta h)\rho_t(z) + \theta h \nu_t(z)} \leq \frac{1}{(1 - \theta h)\rho_t(z)} \leq \frac{2}{\rho_t(z)}.$$

Hence, for $h$ sufficiently small, we have:

$$\|F_h(z;t)\|_2 \leq \left\|\frac{\rho_t(z)}{\rho_t^{\theta h}(z)}\left(s_t^\rho(z) - s_t^{\theta h}(z)\right) - \frac{\nu_t(z)}{\rho_t^{\theta h}(z)}\left(s_t^\nu(z) - s_t^{\theta h}(z)\right)\right\|_2$$

$$\leq \frac{2}{\rho_t(z)}\left\|\rho_t(z)\left(s_t^\rho(z) - s_t^{\theta h}(z)\right) - \nu_t(z)\left(s_t^\nu(z) - s_t^{\theta h}(z)\right)\right\|_2.$$

Applying the triangle inequality, we then obtain:

$$\frac{2}{\rho_t(z)}\left\|\rho_t(z)\left(s_t^\rho(z) - s_t^{\theta h}(z)\right) - \nu_t(z)\left(s_t^\nu(z) - s_t^{\theta h}(z)\right)\right\|_2$$

$$\leq \frac{2}{\rho_t(z)}\left(\rho_t(z)\|s_t^\rho(z) - s_t^{\theta h}(z)\|_2 + \nu_t(z)\|s_t^\nu(z) - s_t^{\theta h}(z)\|_2\right).$$

Now, define

$$k_t^\rho(z) := \int w_t(z, x) x \mathrm{d}\rho(x)$$

and similarly for $k_t^\nu(z)$ and $k_t^{\theta h}(z)$. Then Equation **??** tells us that

$$s_t^\rho(z) = \frac{1}{\sigma_t^2}\left(k_t^\rho(z) - z\right),$$

and similar identities hold for the other score functions. Furthermore,

$$\|s_t^\rho(z) - s_t^{\theta h}(z)\|_2 = \frac{1}{\sigma_t^2}\|k_t^\rho(z) - k_t^{\theta h}(z)\|_2 \leq \frac{1}{\sigma_t^2}\left(\|k_t^\rho(z)\|_2 + \|k_t^{\theta h}(z)\|_2\right),$$

where the last line follows from the triangle inequality. Because $k_t^\rho(z)$ is a convex combination of points in the compact support of $\rho$, we can bound $\|k_t^\rho(z)\|_2 \leq D^\rho < +\infty$, where $D^\rho$ is the diameter of the support of $\rho$. Similarly, $\|k_t^\nu(z)\|_2 \leq D^\nu < +\infty$, and because $\mathrm{supp}(\rho^{\theta h}) \subseteq \mathrm{supp}(\rho) \cup \mathrm{supp}(\nu)$, we have $\|k_t^{\theta h}(z)\|_2 \leq D^\rho + D^\nu$. Substituting these bounds into the above and simplifying, we obtain:

$$\|F_h(z;t)\|_2 \leq \frac{2}{\rho_t(z)} \left( \rho_t(z)\|s_t^\rho(z) - s_t^{\theta h}(z)\|_2 + \nu_t(z)\|s_t^\nu(z) - s_t^{\theta h}(z)\|_2 \right)$$

$$\leq \frac{2}{\rho_t(z)} \left( \frac{\rho_t(z)}{\sigma_t^2}(2D^\rho + D^\nu) + \frac{\nu_t(z)}{\sigma_t^2}(2D^\nu + D^\rho) \right)$$

$$=: G(z;t)$$

This function $G(z;t)$ dominates $\|F_h(z;t)\|_2$ uniformly in $h$ for all $z, t$. It remains to show that $G(z;t) \in L^2(\mathbb{R}^d, \rho_t^0)$. To this end, first note that $\rho_t^0 = \rho_t$. Then,

$$\int G(z;t)\mathrm{d}\rho_t^0(z) = \int G(z;t)\mathrm{d}\rho_t(z)$$

$$= \int \frac{2}{\rho_t(z)} \left( \frac{\rho_t(z)}{\sigma_t^2}(2D^\rho + D^\nu) + \frac{\nu_t(z)}{\sigma_t^2}(2D^\nu + D^\rho) \right) \rho_t(z)\mathrm{d}z$$

$$= \frac{2(2D^\rho + D^\nu)}{\sigma_t^2} \underbrace{\int \rho_t(z)\mathrm{d}z}_{=1} + \frac{2(2D^\nu + D^\rho)}{\sigma_t^2} \underbrace{\int \nu_t(z)\mathrm{d}z}_{=1}$$

$$= \frac{6}{\sigma_t^2}(D^\rho + D^\nu)$$

$$< +\infty.$$

This shows that $G(z;t) \in L^2(\mathbb{R}^d, \rho_t^0)$. As the hypotheses of the DCT are satisfied, we finally conclude that $g_t$ is the Fréchet derivative of $T_t$ at $\eta = 0$ for any $t \in [t_0, t_1]$. This completes the proof of Theorem 3.1. ∎

## C   MOST AND LEAST INFLUENTIAL SAMPLES

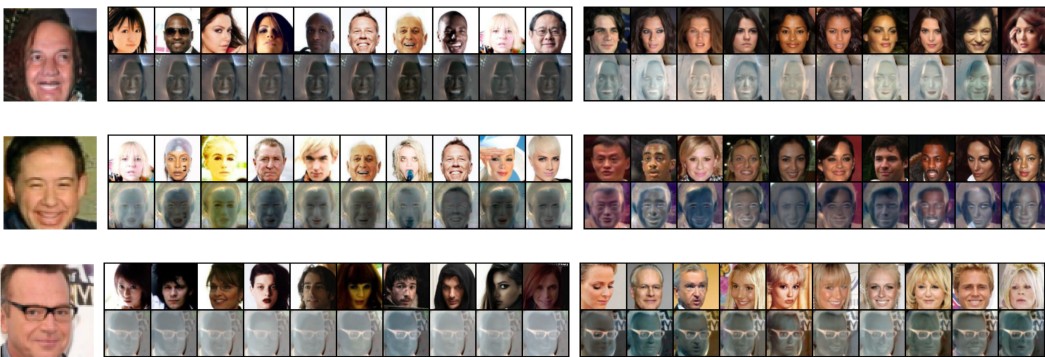

Figure 12: Most and least influential training samples (center, right, resp.) for the model sample on the left, with the corresponding sensitivities on the bottom row.

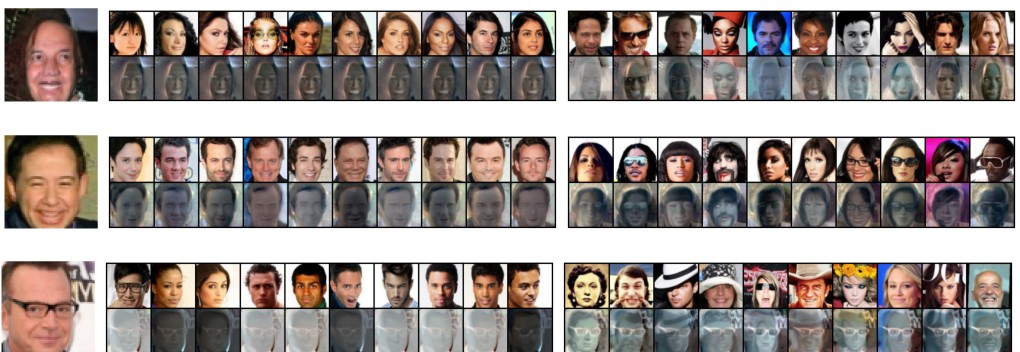

Figure 13: Training samples with the largest and smallest residual influence scores (center, right, resp.) for the model sample on the left, with the corresponding sensitivities on the bottom row.

## D   SINGULAR VECTORS OF SAMPLE SENSITIVITIES

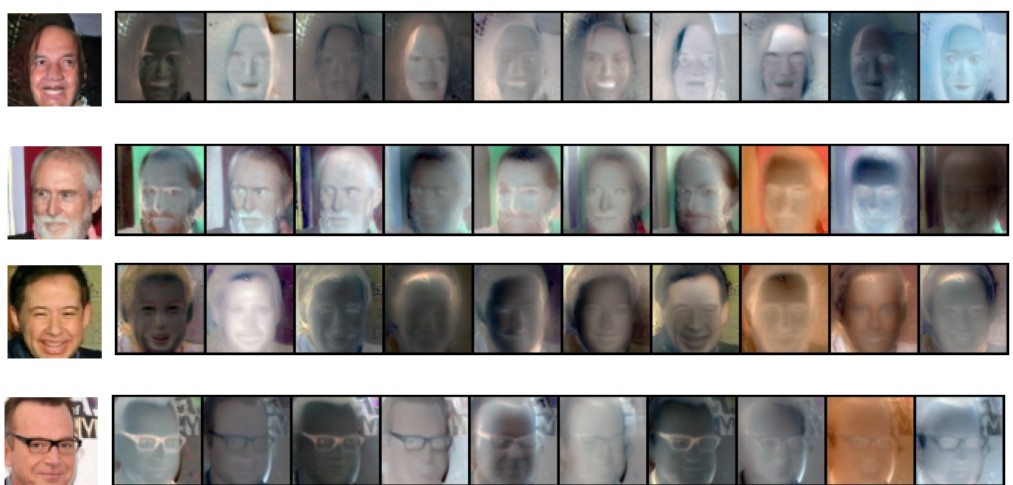

Figure 14: Top 10 right-singular vectors of the $N \times d$ matrix formed from the sensitivities of a single model sample (left) to each of its $N$ training samples. These are interpretable directions in image space; see Figure 10 for examples of perturbing a model sample along several singular directions.

# E  VISUALIZING THE SAMPLE SENSITIVITIES

In this appendix, we illustrate our sample sensitivity analysis on images from the CelebA dataset. We draw four samples from a base model trained on CelebA and solve Equation 2 for each model sample and for four perturbation measures $\nu$, each of which is an empirical measure over a perturbation set $S$. These perturbation sets consist of samples from the CelebA test set possessing the attribute labels "bald", "goatee", "smiling", and "eyeglasses", respectively. We depict model samples in the top row of Figure 15 and solutions to the sample sensitivity ODE for each perturbation set in the bottom four rows. Solutions to this ODE should approximate changes in the model samples in the top row in response to perturbing the base model's target distribution, and many of these predictions are intuitively reasonable in practice. For instance, base model samples representing people without glasses are pushed towards samples of people with glasses in response to perturbing the target distribution towards CelebA samples with the "eyeglasses" attribute, and one observes similar phenomena for the other perturbation sets.

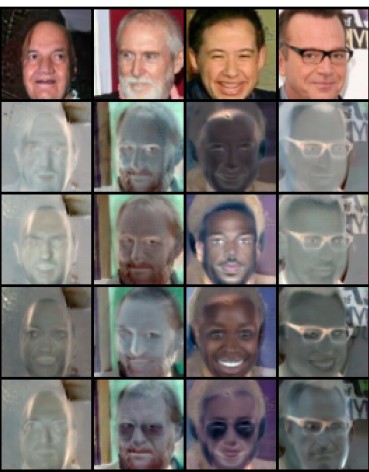

Figure 15: The bottom four rows depict solutions $\left.\frac{\mathrm{d}}{\mathrm{d}\eta}\Phi^{\eta}_{\tilde{t}_1}(z_0)\right|_{\eta=0}$ to the sample sensitivity equation (2) for model samples $\Phi^0_{\tilde{t}_1}(z_0)$ pictured in the top row. In each of the lower four rows, the perturbation measure $\nu$ is the empirical distribution over images from the CelebA test set with attributes "bald", "goatee", "smiling", and "eyeglasses", respectively.

In Figure 16, we also depict line segments of the form $\Phi^0_{\tilde{t}_1}(z_0) + \alpha\left.\frac{\mathrm{d}}{\mathrm{d}\eta}\Phi^{\eta}_{\tilde{t}_1}(z_0)\right|_{\eta=0}$ for $\alpha \in [-2, 2]$ and for the sample sensitivity ODE solutions depicted in Figure 15. These line segments should approximate samples from a model whose target distributed has been perturbed towards $\pm\nu$, where $\nu$ is the empirical measure over CelebA test images with the specified attributes. For $\alpha$ close to 0, the perturbed samples resemble the original sample (6th from the left in each row), differing mainly in the strength of the specified attribute. As $\alpha$ moves farther from 0, the perturbed samples deviate increasingly from the original.

# F  EXPERIMENT DETAILS

## F.1  SYNTHETIC EXPERIMENTS

### F.1.1  FIRST-ORDER APPROXIMATION FOR PERTURBED MODEL SAMPLES

In this experiment, the initial target measure $\rho$ is an equally-weighted mixture of two Gaussians on $\mathbb{R}^{100}$ with means $(-1, ..., -1)$ and $(1, ..., 1)$, respectively, and shared covariance $\sigma^2 I$ for $\sigma = 0.1$. We perturb $\rho$ in the direction of a Gaussian distribution $\nu$ centred at $(1, ..., 1)$ with covariance $\sigma^2 I$ for $\sigma = 0.1$. For any $\bar{\eta} \in [0, 1]$, the perturbed target $\rho^{\bar{\eta}} = (1 - \bar{\eta})\rho + \bar{\eta}\nu$ is a mixture of Gaussians with the same means and covariances as $\rho$, but with weights $\frac{1-\bar{\eta}}{2}$ and $\frac{1+\bar{\eta}}{2}$.

We obtain sample paths $z_t$ for $\rho_t$ and $\rho_t^{\bar{\eta}}$ by fixing 1000 base samples $z_0 \sim \rho_0$ and numerically integrating the PF-ODE and variance-preserving SDE using a forward Euler scheme and Euler-Maruyama scheme, resp., with step sizes $\Delta t \in \{1 \times 10^{-4}, 5 \times 10^{-4}, 1 \times 10^{-3}, 5 \times 10^{-3}\}$. Our scale and noise schedules come from a linear `DDPMScheduler` from the `diffusers` library with $\beta_{\text{start}} = 10^{-4}$ and $\beta_{\text{end}} = 0.02$. We exactly compute the Gaussian mixture densities $\rho_t(z_t)$ along each sample path. We then integrate Equation 2 using the same forward Euler scheme to obtain the sensitivities $\frac{d}{d\eta}\Phi_{\tilde{t}_1}^{\eta}(z_0)\big|_{\eta=0}$ of samples from $\rho_{\tilde{t}_1}$. We compute the Taylor remainder:

$$R(\bar{\eta}) := \left(\Phi_{\tilde{t}_1}^{\bar{\eta}}(z_0) - \Phi_{\tilde{t}_1}^{0}(z_0)\right) - \bar{\eta}\frac{d}{d\eta}\Phi_{\tilde{t}_1}^{\eta}(z_0)\Big|_{\eta=0}$$

and report the median value of $\frac{R(\bar{\eta})}{\bar{\eta}}$ across the 1000 batch samples in our plots.

In our experiments studying the effect of using Hutchinson's estimator to estimate model densities, we use the same setup as in the step size experiments, but estimate the base model densities $\rho_t(z)$ with Hutchinson's estimator: $\text{tr}(A) = \mathbb{E}[\epsilon^\top A \epsilon]$. We use standard normal Gaussian samples for $\epsilon$ and report the number of noise samples we used in our plots.

### F.1.2 STABILITY UNDER SCORE APPROXIMATION ERROR

Here, the initial target measure $\rho$ is an equally-weighted mixture of two Gaussians on $\mathbb{R}^{10}$ with means $(-1, ..., -1)$ and $(1, ..., 1)$ and shared covariance $\sigma^2 I$ for $\sigma = 0.1$. We perturb $\rho$ in the direction of a Gaussian distribution $\nu$ centred at $(1, ..., 1)$ with covariance $\sigma^2 I$ for $\sigma = 0.1$. Instead of evaluating the score of $\rho_t$ in closed form as in 4.1, we now train a neural network to approximate this score function. Our neural network is a two-hidden-layer MLP with SiLU activations and 512-dimensional hidden layers. We also use Fourier features (Tancik et al., 2020) with 128 frequencies and $\sigma = 2.0$. We solve the score-matching problem using AdamW with a learning rate of $10^{-4}$ and a batch size of 100k. We train for 200k steps in total. In our plot, we omit the first two measurements of the correlations for clarity, as the training loss was large and network was very far from convergence during this phase of training.

We fix 1000 base samples $z_0 \sim \rho_0$ and evaluate the sensitivity of model samples from the exact diffusion model $\rho_t$ and its neural approximation every 1000 training steps. We discretize all ODEs using a forward Euler scheme with step size $10^{-2}$ and use Hutchinson's estimator with 100 samples to estimate the model densities $\rho_t(z)$. We measure the median correlation between the exact and approximate sample sensitivities and compare it to the value of the score-matching loss at that training step in Figure 4.

**Computing correlation coefficients.** In Sections 4.2 and 4.3, we measure correlations either between pairs of exact and approximate sample sensitivities, or between our sample sensitivities and differences in model samples post- and pre-perturbation of the training set. In each case, we are interested in the correlation between two tensors of shape $(C, H, W)$, where $C$ is the number of channels and $H, W$ are the height and width, respectively, of image samples generated by the diffusion model. To compute these correlations, we flatten each tensor so that it has shape $(CHW, )$ and use `numpy.corrcoef` to compute the *correlation coefficient* between the pair of vectors. Given two vectors $u, v \in \mathbb{R}^d$, their correlation is computed as follows:

$$\text{Corr}(u, v) := \frac{\langle u - \bar{u}, v - \bar{v} \rangle}{\|u - \bar{u}\|_2 \|v - \bar{v}\|_2},$$

where $\bar{u} := \frac{1}{d}\sum_{i=1}^{d} u_i$ is the mean of $u$ and $\bar{v}$ is defined similarly. This is the cosine similarity between $u$ and $v$ after centering. In the setting of Section 4.3, it measures the extent to which our sample sensitivities can predict increases or decreases in pixel intensity across model samples after retraining or fine-tuning on a perturbed training set.

### F.2 IMAGE DATASETS

**Retraining experiments.** Each neural diffusion model in these experiments is parametrized by a `Unet2DModel` from the `diffusers` library. For the CelebA experiments,

we set `layers_per_block=2`, `block_out_channels=(128, 256, 512, 512)`, and `norm_num_groups=32`. We use a `DDPMScheduler` with $\beta_{\text{start}} = 10^{-4}$ and $\beta_{\text{end}} = 0.02$. The base model samples consist of 10k iid samples from the CelebA training set, and the new samples $S$ are 495 CelebA training samples with a large CLIP score for "a photo of an old man". We pre-process the training images by center-cropping to a size of $140 \times 140$, then resizing to $64 \times 64$ and normalizing to $[-1, 1]$. We apply random horizontal flips as augmentations in training. We then train the CelebA diffusion models for 1000 epochs with an effective batch size of 512. Our optimizer is AdamW with a learning rate of $10^{-4}$.

For the MNIST experiments, we set `layers_per_block=2`, `block_out_channels=(32, 64, 128)`, and `norm_num_groups=8`. We use a `DDPMScheduler` with $\beta_{\text{start}} = 10^{-4}$ and $\beta_{\text{end}} = 0.02$. We do not apply any preprocessing to these samples. We train the MNIST diffusion models for 100 epochs with an effective batch size of 1024. Our optimizer is AdamW with a learning rate of $10^{-4}$.

We draw model samples by integrating the PF-ODE and estimate model densities along the sample path using Hutchinson's estimator with 1 sample. We numerically integrate the PF-ODE and our sample sensitivity ODE (2) using a forward Euler scheme with a step size of $10^{-3}$. We clamp the $\frac{\nu_t(z)}{\rho_t(z)}$ weights to $[0.1, 10]$ for numerical stability. For the entropic OT baseline, we use the `sinkhorn_log` algorithm from the `POT` package (Flamary et al., 2024) with a regularization value of 0.05 to compute the coupling matrix.

**Fine-tuning experiments** These experiments mostly replicate the setup in our retraining experiments, but implement the following changes. For CelebA, we train the base model on 10k iid samples from the CelebA training set for 1k epochs with the same hyperparameters as in the retraining experiments, and then fine-tune for 200 epochs on 495 CelebA training samples with a large CLIP score for "a photo of an old man". We use the same learning rate of $10^{-4}$ for fine-tuning.

For MNIST, we train the base model on the MNIST training set for 100 epochs with an effective batch size of 1024 and a learning rate of $10^{-4}$, and then fine-tune on TMNIST for a single epoch at a learning rate of $10^{-5}$.

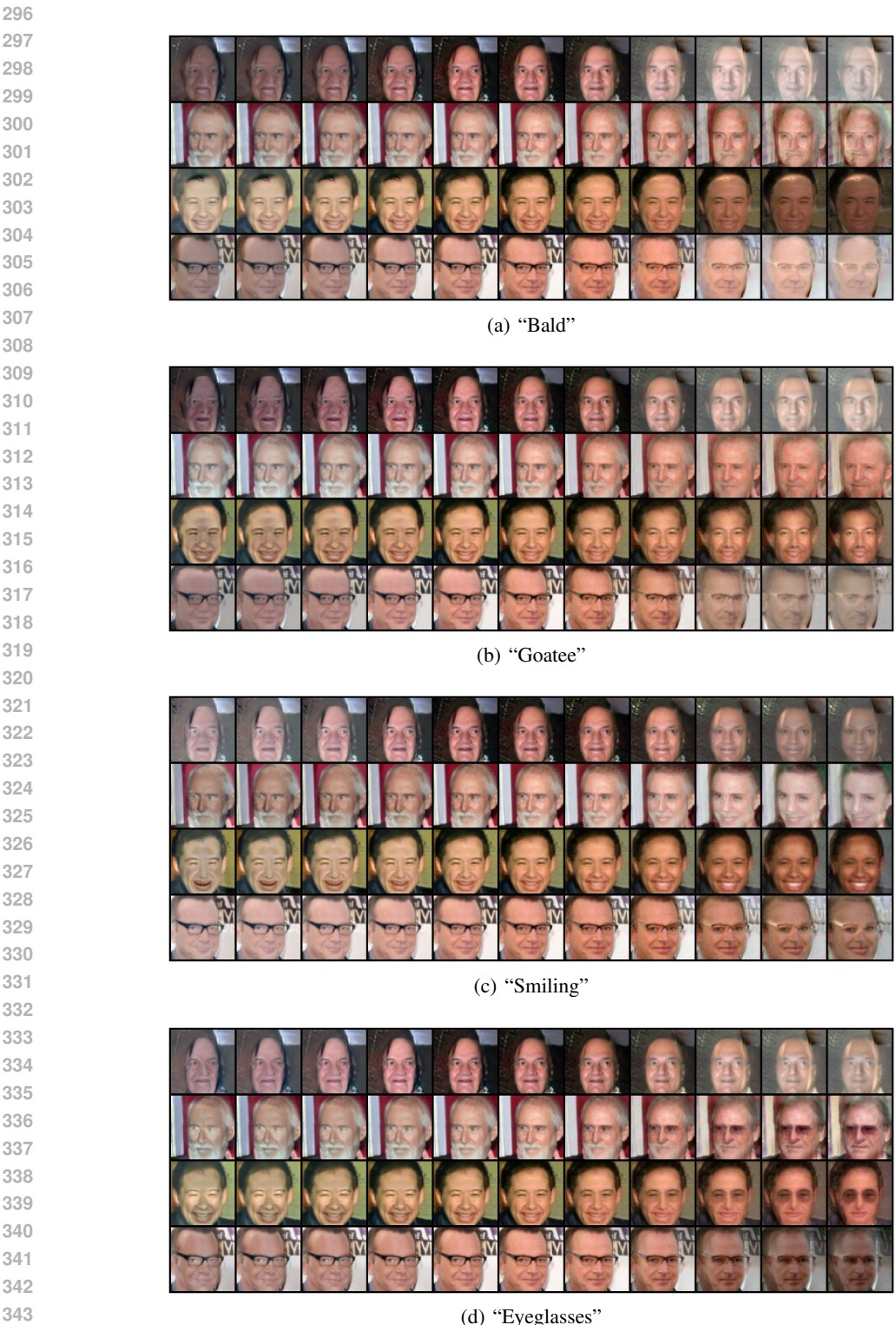

(a) "Bald"

(b) "Goatee"

(c) "Smiling"

(d) "Eyeglasses"

Figure 16: Line segments extending from model samples (center images) towards negative (left) and positive (right) multiples of sample sensitivities $\left.\frac{\mathrm{d}}{\mathrm{d}\eta}\Phi^{\eta}_{t_1}(z_0)\right|_{\eta=0}$. In each subfigure, the perturbation measure $\nu$ is the empirical distribution over CelebA test samples with the attribute listed in the subcaption.

