# OpenReview forum: "Sensitivity Analysis for Diffusion Models"
_ICLR.cc/2026/Conference — Submitted to ICLR 2026_

### Official Review · Reviewer_xkKn · 2025-10-28

**Soundness:** 3
**Presentation:** 2
**Contribution:** 2
**Rating:** 4
**Confidence:** 3

**Summary:**

This article introduces a method for computing the sensitivity of diffusion models, modeled as the directional derivative of both the learned scores and the generated samples. The computation requires only the pre-trained score estimator and its spatial derivatives. Experiments on synthetic and real-world datasets validate the method's effectiveness.

**Strengths:**

- **Originality**: The core idea of quantifying the influence of training distribution perturbations is novel.

- **Quality**: The motivation for the sensitivity calculation is well-justified, and the mathematical deductions are both clear and sound.

- **Clarity**: The paper is well-structured; the theory is presented logically and is easy to follow.

- **Significance**: Sensitivity analysis is highly significant for ensuring the safety and reliability of diffusion models in real-world applications.

**Weaknesses:**

- The practical use of this sensitivity study is not well presented.
- The numerical comparison is limited. In particular, the baseline method used for the real-world data experiments (from 2013) is outdated and does not convincingly demonstrate superiority over modern techniques.
- Some descriptions are not necessary. For example:
  - Equation 1, which formulates the optimal score, is not strongly correlated with the core content and could be removed to improve focus.
  - In Sec. 3.3, there is no need to illustrate the reason for choosing $\tilde{t}_{1} < t_{1}$ since it is not strongly correlated to the content.
  - For explaining how to compute the perturbation in score and samples, an algorithm could be more illustrative than just sentences in Sec. 3.3 and Sec. 4.1.

**Questions:**

- What is the application scenario of this study? How does it apply to the safety of deploying diffusion models?
- In Figure 4, why does the first-order approximation seem to fail for $\bar{\eta} = 10^{-6}$?

---

> ### Author Response · Authors · 2025-11-23
>
> Thank you for your thoughtful review. We appreciate your recognition of the novelty of our approach, the clarity and soundness of our theory, and the broader significance of a sensitivity analysis framework for diffusion models. **We have uploaded a revised manuscript in which we address many of your concerns**, and provide further details below.
>
> *The numerical comparison is limited. In particular, the baseline method used for the real-world data experiments (from 2013) is outdated and does not convincingly demonstrate superiority over modern techniques.*
>
> This critique is inaccurate: Our method does not use a baseline method from 2013. We believe you are referring to “Sinkhorn distances: Lightspeed computation of optimal transport” by Cuturi (2013). This is simply a reference to the work that popularized entropy-regularized optimal transport (entropic OT) in machine learning. Entropic OT remains a standard technique for efficiently approximating optimal transport (OT) couplings and distances in large-scale problems. We use entropic OT to approximate an optimal coupling between samples from the original model and samples from the perturbed model’s target distribution, and use line segments between coupled samples as baseline predictions of how the base model’s outputs may respond to perturbations of its training set. We provide further details on this baseline in Section 4.3 of the revised manuscript and background on entropy-regularized OT in a new Appendix A: Extended Preliminaries.
>
> *Equation 1, which formulates the optimal score, is not strongly correlated with the core content and could be removed to improve focus.*
>
> As several reviewers have agreed that Eq 1 is unnecessary for our exposition, we have removed it from the revised manuscript.
>
> *In Sec. 3.3, there is no need to illustrate the reason for choosing $\tilde{t}{1} < t{1}$ since it is not strongly correlated to the content.*
>
> We must truncate the solution interval to ensure that the velocity field remains sufficiently regular to apply the results from Khalil (2002) and Kunita (2019).
>
> *For explaining how to compute the perturbation in score and samples, an algorithm could be more illustrative than just sentences in Sec. 3.3 and Sec. 4.1.*
>
> Thank you for this suggestion. Algorithm 1 in the revised manuscript now summarizes our sample sensitivity analysis from Section 3.3, including the computation of the sample paths and log-densities for both $\rho_t$ and $\nu_t$. Using these sample paths and log-densities, one may immediately compute the score sensitivities using Eq 1, which are in turn used in the sample sensitivity ODE specified in Eq 2.
>
> *What is the application scenario of this study?*
>
> Our primary contribution is a sensitivity analysis framework for diffusion models, which allows us to predict how a model’s score function and generated samples change under perturbations of the training set. To the best of our knowledge, ours is the first framework that can predict how a diffusion model's samples change under small perturbations to the training set with rigorous theoretical convergence guarantees for optimal diffusion models. We outline several promising directions for future work in Section 5 of the manuscript.
>
> *In Figure 4, why does the first-order approximation seem to fail for $\bar{\eta} = 10^{-6}$?*
>
> Because the model samples from the original and perturbed models are computed by numerically integrating an ODE, there is an intrinsic noise floor on the Taylor remainder $R(\bar{\eta})$ in this experiment. This causes the little-o error $\frac{R(\eta)}{\eta}$ to rise for small values of $\eta$.
>
> **Conclusion.**
>
> Thank you for your thoughtful review and your helpful feedback on our work. **If we have addressed your concerns, we respectfully ask that you consider raising your score for our submission.** Otherwise, we would be happy to continue this conversation during the discussion period.

---

### Official Review · Reviewer_WTVj · 2025-10-30

**Soundness:** 2
**Presentation:** 2
**Contribution:** 3
**Rating:** 4
**Confidence:** 3

**Summary:**

This work derives closed-form directional derivatives of score functions w.r.t. data distributions given another external distribution, which characterizes certain input sensitivity of diffusion models. The results are also extended to the solution path case (sensitivity ODEs) when sampling (with probability flow ODEs or certain SDEs). This helps to predict changes in model samples after retraining and fine-tuning.

**Strengths:**

1. The paper is well-written and clearly organized, which is fluent to read.
2. The sensitivity problem of diffusion models is novel, and quite different from that of general neural networks due to the dynamic formulation of diffusion models. This work takes an initial step towards this direction.
3. The derived theoretical result is clean and insightful.
4. Numerical verifications are consistent, with meticulous robustness studies covering main components of the proposed calculation method.

**Weaknesses:**

1. It would be clearer to add a dedicated algorithm of the proposed calculation method (e.g. Thm. 3.1).
2. Following 1, how can we compute score functions *accurately* in Eq. (2) in detail? How can we compute probability densities in Eq. (2) *efficiently* beyond neural ODEs? It would be better to include self-contained algorithms regarding them.
3. Following 1, I also suggest to formulate in-paragraph discussions in Sec. 3.3 as separate algorithms.
4. For Eq. (1): It is just the score function of $Z_t$, right? What is the meaning of "optimal solution to this problem (score-matching)" (Line 107)?
5. What are the computation and memory complexity of Eq. (2) and Eq. (3) w.r.t. time steps & data dimensions?
6. It would be more readable to provide self-contained supports in former references for key quantities (at least in appendices), e.g. CCoV, sensitivity equations, and entropic optimal transport (OT) coupling.
7. Experiments are mainly conducted on MNIST and CelebA datasets. How about the performance on standard Cifar (more diversity) and the efficiency on large-scale ImageNet datasets?
8. Can authors provide more detailed instructions on how the correlation is calculated in e.g. Fig. 6?
- Although the proposed calculation method outperforms OT (baseline), it is still far away from 1. What are potential sources of this gap?
- It is not clear why the correlation is better for larger datasets like CelebA instead of smaller MNIST?

**Questions:**

See weaknesses.

---

> ### Author Response · Authors · 2025-11-23
> **Rebuttal to Reviewer WTVj, Part 1/2**
>
> Thank you for your thoughtful review. We are grateful for your recognition of our theoretical contributions and the thoroughness of our robustness study. **We have uploaded a revised manuscript in which we address many of your concerns**, and provide further details below.
>
> *It would be clearer to add a dedicated algorithm of the proposed calculation method (e.g. Thm. 3.1). [...] Following 1, I also suggest to formulate in-paragraph discussions in Sec. 3.3 as separate algorithms.*
>
> Thank you for this suggestion. Algorithm 1 in the revised manuscript now summarizes our sample sensitivity analysis from Section 3.3, including the computation of the sample paths and log-densities for both $\rho_t$ and $\nu_t$. Using these sample paths and log-densities, one may immediately compute the score sensitivities using Eq 1, which are in turn used in the sample sensitivity ODE specified in Eq 2.
>
> *Following 1, how can we compute score functions accurately in Eq. (2) in detail? How can we compute probability densities in Eq. (2) efficiently beyond neural ODEs? It would be better to include self-contained algorithms regarding them.*
>
> In practice, we typically access the base diffusion model $\rho_t$ through a neural network parametrizing its score function $s^\rho_t$. While we cannot directly access the density of $\rho_t$ under this parametrization, the continuous change of variables (CCoV) formula suggests an algorithm for computing densities along a diffusion model’s sample path. We provide further details on the CCoV formula in Appendix A: Extended Preliminaries, which is a new addition to our revised manuscript.
>
> In our setting, the perturbation measure $\nu$ is often the empirical measure over $K$ samples $x_k$ that we would like to add or remove from the training set. In this case, $\nu_t$ is a mixture of isotropic Gaussians with means $\alpha_t x_k$ and common covariance $\sigma_t^2 I$, where $\alpha_t$ and $\sigma_t$ denote the scaling and noise schedules, respectively. One can compute the exact score and density of $\nu_t$ in $O(dK)$ time, where $d$ is the data dimension, and in settings where $K$ is large enough to render this prohibitive, one may use techniques from Karppa et al. (2022) and Scarvelis et al. (2025) to approximate these functions using approximate nearest-neighbor queries. Appendix A also provides further details on these computations. We elaborate on these points in the discussion following Theorem 3.1.
>
> *For Eq. (1): It is just the score function of $Z_t$, right? What is the meaning of "optimal solution to this problem (score-matching)" (Line 107)?*
>
> This is correct: The optimal solution to the score-matching problem is the score of the random variable $Z_t$. As several other reviewers agreed that Eq 1 was superfluous to the paper, we have removed it from the revised manuscript.
>
> *What are the computation and memory complexity of Eq. (2) and Eq. (3) w.r.t. time steps & data dimensions?*
>
> We have included a big-O estimate of our sample sensitivity analysis’ time and memory complexity in the paragraph titled “ODE sampling” in the revised manuscript. Our sample sensitivity analysis requires $O(hT)$ evaluations of $s_t^\rho$, where $h$ is the number of samples used in Hutchinson’s estimator and $T$ is the number of time steps; $O(T)$ evaluations of $s_t^\nu$; and $O(T)$ evaluations of $\nu_t$. In all experiments in our paper, $s^\rho_t$ is parametrized by a neural network, whose inference time complexity is architecture-dependent, and $\nu$ is the empirical distribution over $K$ samples. In this case, $s_t^\nu, \nu_t$ are the score and density, respectively, of a mixture of Gaussians, which each take $O(dK)$ time per evaluation, where $d$ is the data dimension. **The total time complexity of our sample sensitivity analysis is $O((hP+dK)T)$**, where P is the architecture-dependent cost of evaluating the score network $s_t^\rho$. **Our sample sensitivity analysis’ memory complexity is $O(dT)$**, where $d$ is the data dimension.
>
> *It would be more readable to provide self-contained supports in former references for key quantities (at least in appendices), e.g. CCoV, sensitivity equations, and entropic optimal transport (OT) coupling.*
>
> Thank you for this suggestion. Appendix A: Extended Preliminaries in the revised manuscript now provides background information on the CCoV formula, entropy-regularized optimal transport, and the ODE sensitivity equation.
>
> *Experiments are mainly conducted on MNIST and CelebA datasets.*
>
> We have added Section 4.4: Empirical Properties of Sample Sensitivities to our revised manuscript, in which we study the statistics of our sample sensitivities and its magnitudes. The paragraph titled “Cross-class sensitivities are small” leverages the availability of class labels for CIFAR-10 to show that cross-class sensitivities are typically small in practice.
>
> **Continued in Part 2/2**

---

> ### Author Response · Authors · 2025-11-23
> **Rebuttal to Reviewer WTVj, Part 2/2**
>
> *Can authors provide more detailed instructions on how the correlation is calculated in e.g. Fig. 6?*
>
> We have added a paragraph titled “Computing correlation coefficients” to Appendix D.1.2 in the revised manuscript, in which we explain in detail how the correlation coefficients are computed throughout our paper.
>
> *Although the proposed calculation method outperforms OT (baseline), it is still far away from 1. What are potential sources of this gap? It is not clear why the correlation is better for larger datasets like CelebA instead of smaller MNIST?*
>
> We have expanded our discussion of these results in Section 4.3 of the revised manuscript. In particular, we explain why we would not expect the correlations to attain values close to 1 and conjecture as to why our method performs better on CelebA than on MNIST (see lines 396-407).
>
> **Conclusion.**
>
> Thank you for your thoughtful review and your helpful feedback on our work. **If we have addressed your concerns, we respectfully ask that you consider raising your score for our submission.** Otherwise, we would be happy to continue this conversation during the discussion period.

---

### Official Review · Reviewer_DS2B · 2025-10-31

**Soundness:** 2
**Presentation:** 2
**Contribution:** 2
**Rating:** 4
**Confidence:** 3

**Summary:**

This paper studies the differentiablity of the diffusion model training map, that is the map that assigns a score function to a dataset.
An algorithm is proposed, which allow studying the sensitivity of the diffusion model to addition or removal of dataset points (images).
This is then exploited to define sample sensitivity.

**Strengths:**

* The main strength of the paper is to establish Theorem 3.1.
* Original problem

**Weaknesses:**

I don't see where the term "sample sensitivity" is clearly defined? I understand that this is the derivative involved in Eq (4) as confirmed by Fig. 10 caption, but this is not clear from the text.

Theorem 3.1 suppose bounding support but all examples use mixture of Gaussians. A comment on that point seems necessary.

**Experiments:**
* Section 4.1: Why use $d=100$ with mixture of isotropic Gaussians. This looks like to poor a model.
* Section 4.2: As explained in Section C.1.2, here the work is done in $d=10$ with a bimodal Gaussian mixture. This is a very simple model to learn with a network.
* Sections 4.2 and 4.3: Why is it enough to measure the experiment by only requiring better correlations? How is this computation stable since for most images/points the output should be very close to zero?
* Figure 6: The OT baseline is based on empirical OT between samples. This could be replaced by a parameterized transport map trained on the whole dataset (see eg Korotin et al ICLR 2023 and references therein).
* Figure 8: What is displayed in figure 8,

Minor remarks:
* No hyperlink on cross-references

**Questions:**

See questions regarding experiments.

---

> ### Author Response · Authors · 2025-11-23
>
> Thank you for your thoughtful review. We are glad that you appreciate the originality of our sensitivity analysis framework for diffusion models and our theoretical contributions. **We have uploaded a revised manuscript in which we address many of your concerns**, and provide further details below.
>
> *I don't see where the term "sample sensitivity" is clearly defined?*
>
> We now explicitly introduce the term “sample sensitivity” in the paragraph titled “ODE sampling” in Section 3.3 (see line 208).
>
> *Theorem 3.1 suppose bounding support but all examples use mixture of Gaussians. A comment on that point seems necessary.*
>
> Thank you for this observation. We have updated our statement of Theorem 1 to clarify that our score sensitivity formula holds pointwise at any $z \in \mathbb{R}^d$ without any assumptions on the support of $\mu$ or $\nu$. (We assume compact support to lift this pointwise derivative to a Fréchet derivative in function space.) The experiments using mixtures of Gaussians in Section 4 rely only on pointwise derivatives, so the measures in these experiments do not require compact support for our theory to apply.
>
> *Section 4.1: Why use $d=100$ with mixture of isotropic Gaussians. This looks like to poor a model. [...] Section 4.2: As explained in Section C.1.2, here the work is done in $d=10$ with a bimodal Gaussian mixture. This is a very simple model to learn with a network.*
>
> We use mixtures of well-separated Gaussians in this section so that we can evaluate the score and density of the resulting diffusion model in closed form. This allows us to isolate the effects of ODE/SDE discretization error, density approximation error, and score approximation error on our sensitivity analysis, which is our goal in Sections 4.1 and 4.2. Mixtures of well-separated Gaussians exhibit pathologies that make it a non-trivial test case for this purpose: In particular, their score functions are nearly discontinuous at the Voronoi boundaries between mixture means when their variance is small. Our experiments in Section 4.3 then demonstrate our method’s effectiveness on real-world data. We have further clarified these points in the revised manuscript.
>
> *Sections 4.2 and 4.3: Why is it enough to measure the experiment by only requiring better correlations? How is this computation stable since for most images/points the output should be very close to zero?*
>
> In Sections 4.2 and 4.3, we measure correlations either between pairs of exact and approximate sample sensitivities, or between our sample sensitivities and differences in model samples post- and pre-perturbation of the training set. These correlations are cosine similarities between pairs of centered vectors; at another reviewer’s request, we have provided further details on this computation in Appendix F.1.2 in the revised manuscript.
>
> Measuring the accuracy of a theoretical prediction of an image model’s outputs via correlations is standard practice in the literature; see e.g. “An analytic theory of creativity in convolutional diffusion models” by Kamb and Ganguli (2025). In practice, we do not find that the magnitude of our sample sensitivities is small enough to merit concerns about numerical stability. For instance, in Section 4.4 in the revised manuscript, we compute the magnitude of a CelebA’s sample sensitivities wrt each of its training samples to find the top- and bottom-10 influential training samples. In this experiment, the magnitudes of the sample sensitivities range from 10.65 to 121.04.
>
> *Figure 6: The OT baseline is based on empirical OT between samples. This could be replaced by a parameterized transport map trained on the whole dataset (see eg Korotin et al ICLR 2023 and references therein).*
>
> A fair baseline for our sensitivity analysis must be training-free, as a key advantage of our approach is its ability to predict changes in model samples after retraining before committing to a costly training run. Computing Korotin et al.’s neural optimal transport map is roughly as expensive as training a generative model, making it an unsuitable baseline for our method.
>
> *Figure 8: What is displayed in figure 8,*
>
> We have updated this figure (now Figure 7) to include the corresponding model samples pre- and post-perturbation and provided further details on the figure in the revised manuscript.
>
> *No hyperlink on cross-references*
>
> Our references throughout the revised manuscript are hyperlinked and colored in blue and maroon.
>
> **Conclusion.**
>
> Thank you for your thoughtful review and your helpful feedback on our work. **If we have addressed your concerns, we respectfully ask that you consider raising your score for our submission.** Otherwise, we would be happy to continue this conversation during the discussion period.

---

### Official Review · Reviewer_abcK · 2025-10-31

**Soundness:** 4
**Presentation:** 3
**Contribution:** 3
**Rating:** 6
**Confidence:** 4

**Summary:**

Diffusion models and the mathematical objects that define them (the score function and the learned distribution) are functions of training data. This means that, if the model were trained on additional data, or some training example were removed from the training set, these mathematical objects would be different, and hence generated samples would be different. How different would they be? How sensitively does a given diffusion model depend on one of its training examples?

This is the question the authors seek to address in their paper. They formalize the question mathematically in terms of the Frechet derivative of a perturbation map, use this formalization to motivate a specific numerical approach to measuring sensitivity to training data, and then show that their method works in a number of experiments. They also place their work in the context of related machine learning work, e.g., related to influence functions.

**Strengths:**

Overall, I really like this paper. There is a clear goal---How do we formalize and measure the extent to which a diffusion model depends on small perturbations to its training set?---along with math, an algorithmic approach, and experiments to back it up. The paper is also reasonably well-written and clear throughout.

Lots of details seem to have been carefully attended to. I also like how the sequence of figures shown slowly build up the ideas of the paper, and gradually show that the method works.

**Weaknesses:**

I think my overall complaints are minor. One of them is that a lot of space is dedicated to showing that the approach yields sensible results (e.g., Figs. 4-7), but not much space is dedicated to showing what the sensitivity results actually are and what we learn from them. Some SI figures show a bit of this (Figs. 9 and 10); I think the paper would be better if some of these were moved to the main text.

Relatedly, some material currently in the main text could probably be moved to SI. For example, Eq. 1 doesn't seem to be used anywhere, and doesn't say anything interesting. The proof that the method is reasonable is belabored a bit, and maybe some of the details of Sec. 4 could be moved to SI.

The figures could be improved slightly. Many of the figures have small text or labels, and would be improved by making those things bigger (Figs. 4-7 especially). In Figure 8, it would be helpful to also include (i) the perturbation, and (ii) the images pre-perturbation, as opposed to just the sensitivities and post-perturbation images. It's hard to parse the figure without this extra info.

**Questions:**

1. Can the authors say more about the 'optimal transport baseline' in the main text? I didn't understand that part.

2. Are there interesting things worth sharing about what we learn from measuring a bunch of sensitivities? Are certain kinds of data points (e.g., outliers) more influential than others (e.g., non-outliers)?

---

> ### Author Response · Authors · 2025-11-23
>
> Thank you for your thoughtful review. We are glad that you appreciate our sensitivity analysis framework for diffusion models. **We have uploaded a revised manuscript in which we address many of your concerns** and provide further details below.
>
> *One of them is that a lot of space is dedicated to showing that the approach yields sensible results (e.g., Figs. 4-7), but not much space is dedicated to showing what the sensitivity results actually are and what we learn from them. Some SI figures show a bit of this (Figs. 9 and 10); I think the paper would be better if some of these were moved to the main text. Relatedly, some material currently in the main text could probably be moved to SI. For example, Eq. 1 doesn't seem to be used anywhere, and doesn't say anything interesting. The proof that the method is reasonable is belabored a bit, and maybe some of the details of Sec. 4 could be moved to SI.*
>
> Thank you for this valuable feedback. We have removed Equation 1 from the manuscript and compressed our numerical validation in Sections 4.1 and 4.2. We have also added a new section, in which we use our method to compute the sensitivity of a CelebA model’s samples to perturbing the weights of individual training samples and study key statistics of the resulting sample sensitivities. We provide further details in an answer to another question of yours below.
>
> *The figures could be improved slightly. Many of the figures have small text or labels, and would be improved by making those things bigger (Figs. 4-7 especially). In Figure 8, it would be helpful to also include (i) the perturbation, and (ii) the images pre-perturbation, as opposed to just the sensitivities and post-perturbation images. It's hard to parse the figure without this extra info.*
>
> We have increased the font size in Figures 4-7 and included pre-perturbation images and the corresponding samples from the fine-tuned model in Figure 8.
>
> *Can the authors say more about the 'optimal transport baseline' in the main text? I didn't understand that part.*
>
> We have provided more details on the optimal transport baseline in Section 4.3 (lines 372-377), and would be happy to answer any remaining questions in the remainder of the discussion period. In response to another reviewer’s comments, we have also provided some background on entropy-regularized optimal transport in Appendix A of the revised manuscript.
>
> *Are there interesting things worth sharing about what we learn from measuring a bunch of sensitivities? Are certain kinds of data points (e.g., outliers) more influential than others (e.g., non-outliers)?*
>
> The new section in our revised manuscript (Section 4.4: Empirical properties of sample sensitivities) studies the statistics of our sample sensitivities for popular image datasets such as CelebA and CIFAR-10. We find that outliers in the distribution of sample sensitivity magnitudes tend to be determined by large-scale features of the training images, such as hair color or background color, and that the magnitude of a model sample’s sensitivity to a training sample is correlated with the L2 distance between the model sample and the training sample. Moreover, any given model sample’s sensitivities with respect to each of its training samples lie mostly in a very low-dimensional subspace, with 93% of the variance in the sample sensitivities in our experiment explained by their top 10 singular vectors. These singular vectors yield interpretable perturbations to the model sample, which we depict in Figure 10 of the revised manuscript. We also exploit the availability of class labels in CIFAR-10 to compare the sensitivity of model samples across classes and find that cross-class sensitivities are small for CIFAR-10.
>
> **Conclusion.**
>
> Thank you for your thoughtful review and your positive feedback on our work. **If we have addressed your concerns, we respectfully ask that you consider raising your score for our submission.** Otherwise, we would be happy to continue this conversation during the discussion period.

---

### Meta-Review · Area_Chair_cjTu · 2026-01-06

**Summary:**

This paper receives 6444 initial ratings and no reviewer replied during the discussion, making the decision difficult.

After reading the manuscript and all the discussion, the AC has two main concerns: (1) The four reviewers raise many questions regarding the clarity, organization, experiment setting, result presentation, method details, which require many changes in the manuscript and may need further review. (2) The practical value of this research is not presented clearly. Though during rebuttal the author add some outlook for possible future applications, it is better to add some real applications to the experiment because it is not direct to bridge this sensitivity analysis method with interpreting CFG, curating the training set  and generalization in diffusion models.


Based on above two concerns, the AC thinks the manuscript is not ready for publication, though the proposed method is very interersting. It will improve the contribution significantly if the proposed method can be applied to interpreting CFG, curating the training set or generalization in diffusion models. It is suggested to add some experiments in future versions.

**Reviewer Concerns:**

Solved:

Reorganization of the main text and SI to show what the sensitivity results actually are and what we learn from them.

Many of the figures have small text or labels, and would be improved by making those things bigger.

Where is the term "sample sensitivity" defined?

Theorem 3.1 suppose bounding support but all examples use mixture of Gaussians.

Why use d=100 with mixture of isotropic Gaussians.

Why is it enough to measure the experiment by only requiring better correlations? How is this computation stable since for most images/points the output should be very close to zero?

What is displayed in figure 8.

Add a dedicated algorithm of the proposed calculation method.

Include self-contained algorithms regarding coputing score functions and probability densities.

Many other detailed questions regarding method clarity and experiments.

still outstanding concerns:

The practical use of this sensitivity study is not well presented.    (xkKn)

**Reviewer Scores:**

abcK may keep 6.
DS2B may keep 4 or increase.
WTVj may keep 4 or increase.
xkKn may keep 4 or increase.

---

### Decision · Program_Chairs · 2026-01-26

Reject